

# All aboard! Earth system investigations with the CH2O-CHOO TRAIN v1.0

Tyler Kukla[1], Daniel E. Ibarra[2,3], Kimberly V. Lau[4], and Jeremy K.C. Rugenstein[1,5]

[1]Department of Geosciences, Colorado State University, Fort Collins, CO, USA
[2]Department of Earth, Environmental and Planetary Sciences, Brown University, Providence, RI, USA
[3]Institute at Brown for Environment and Society, Brown University, Providence, RI, USA
[4]Department of Geosciences and Earth and Environmental Systems Institute, The Pennsylvania State University, University Park, PA, USA
[5]Max Planck Institute for Meteorology, Hamburg, Germany

**Correspondence:** Jeremy K.C. Rugenstein (Jeremy.Rugenstein@colostate.edu); Tyler Kukla (tykukla@colostate.edu)

**Abstract.** Models of the carbon cycle and climate on geologic ($> 10^4$ year) timescales have improved tremendously in the last 50 years due to parallel advances in our understanding of the Earth system and the increase in computing power to simulate its key processes. Despite these advances, balancing the Earth System's vast complexity with a model's computational expense is a primary challenge in model development. Running longer simulations spanning hundreds of thousands of years or more

generally requires reducing the complexity of the modeled climate system. However, simpler model frameworks often leave out certain features of the climate system, such as radiative feedbacks, shifts in atmospheric circulation, and the expansion and decay of ice sheets, which can have profound effects on the long-term carbon cycle. Here, we present a model for climate and the long-term carbon cycle that captures many fundamental features of global climate while retaining the computational efficiency needed to simulate millions of years of time. The Carbon-$H_2O$ Coupled HydrOlOgical model with Terrestrial

Runoff And INsolation, or CH2O-CHOO TRAIN, couples a one-dimensional (latitudinal) moist static energy balance model of climate with a model for rock weathering and the long-term carbon cycle. The key advantages of this framework are (1) it simulates fundamental climate forcings and feedbacks; (2) it accounts for geographic configuration; and (3) it is highly customizable, equipped to easily add features, change the strength of feedbacks, and prescribe conditions that are often hard-coded or emergent properties of more complex models, such as climate sensitivity and the strength of meridional heat transport.

The CH2O-CHOO TRAIN is capable of running million-year-long simulations in about thirty minutes on a laptop PC. This paper outlines the model equations, presents a sensitivity analysis of the climate responses to varied climatic and carbon cycle perturbations, and discusses potential applications and next stops for the CH2O-CHOO TRAIN.

## 1   INTRODUCTION

Interactions between the long-term carbon cycle and global climate govern the habitability of our planet. These interactions are

mediated by complex relationships with factors such as geography, lithology, climate feedbacks, and more (Bluth and Kump, 1994; Caves et al., 2016; Donnadieu et al., 2006; Gibbs and Kump, 1994; Jellinek et al., 2020; Park et al., 2020). Over the last



50 years, a suite of models ranging in complexity have been developed to explore these interactions, with each model carrying its own advantages and drawbacks (Arndt et al., 2011; Bergman, 2004; Berner, 1991, 2004; Colbourn et al., 2013; Donnadieu et al., 2004, 2006; Francois and Walker, 1992; Goddéris and Joachimski, 2004; Kump and Arthur, 1999; Lenton et al., 2018; Mills et al., 2017; Ozaki and Tajika, 2013; Ridgwell et al., 2007; Zeebe, 2012).

One major challenge in building these models is balancing the complexity of the global climate system with the computational efficiency needed to simulate thousands to millions or billions of years of time. Based on the model's intended applications, different frameworks address this trade-off in different ways. Lower-dimensional box models, for example, tend to distill global climate down to a few simple parameters (and in many cases, a single forcing variable, $pCO_2$), usually opting to ignore many factors such as geography, orbital forcing, and ice sheet dynamics (Berner, 1991; Bergman, 2004; Caves et al., 2016; Kump and Arthur, 1997; Lenton et al., 2018; Zeebe, 2012). The simpler representation of climate makes these models highly efficient while leaving room for more complex representations of other factors, such as sedimentary reservoirs and ocean biogeochemical cycling (Zeebe, 2012; Ozaki and Tajika, 2013). Higher-dimensional models, in contrast, capture more complexity in global climate and generally provide the most physically realistic representations of the Earth System on long timescales (Baum et al., 2022; Donnadieu et al., 2006; Holden et al., 2016; Otto-Bliesner, 1995; Ridgwell et al., 2007). However, these models are more computationally expensive, making it harder to efficiently explore the large, multi-dimensional parameter space of its simulated climate system.

The goal of this work is to build a model that remains computationally efficient while capturing features of climate that are usually reserved for more computationally expensive models. This model, the CH2O-CHOO TRAIN (Carbon-$H_2O$ Coupled HydrOlOgical model with Terrestrial Runoff And INsolation) considers factors such as the spatial pattern of radiative climate feedbacks, geography, lithology, insolation, hydroclimate, and more. The model framework couples a moist static energy balance model of climate (Flannery, 1984; Roe et al., 2015; Siler et al., 2018) with a continental weathering model (Maher and Chamberlain, 2014; Winnick and Maher, 2018) and box model for the long-term carbon cycle (Caves Rugenstein et al., 2019; Shields and Mills, 2017). The model is designed to be highly customizable, making it easy to directly modify processes in the climate system such as the strength of climate feedbacks, the sensitivity of runoff, the strength of atmospheric poleward energy transport, and the role of ice sheets in climate and weathering. Such processes have complex interactions with the global carbon cycle that are often highly parameterized or absent from lower dimensional models. Conversely, in higher dimensional models, these processes—particularly atmospheric energy transport and the pattern of certain feedbacks—are often emergent properties, not inputs that can be directly modified. Thus, the CH2O-CHOO TRAIN framework makes it possible to explore how many aspects of climate, especially the water and carbon cycles, interact over space and time across millions of years.




The key feature that allows the CH2O-CHOO TRAIN to run efficiently is the one-dimensional (latitudinal) moist energy balance climate model (MEBM) (Flannery, 1984; Frierson et al., 2006; Hill et al., 2022; Roe et al., 2015; Siler et al., 2018). Energy balance climate models have previously been used with models of the long-term carbon cycle in an effort to efficiently simulate climate without compromising too much complexity. Zero-dimensional global mean energy balance models have been coupled to carbon cycle and weathering models to probe how climate and weathering impact planetary habitability (Abbot et al., 2012; Graham and Pierrehumbert, 2020). A two-dimensional (latitude and longitude) moist energy balance model for the land






and atmosphere has been used in the cGENIE framework (Edwards and Marsh, 2005; Marsh et al., 2011; Ridgwell et al., 2007), retaining a great deal of spatial complexity without having to run the climate model "offline", as is common when more complex climate models are used (Baum et al., 2022; Donnadieu et al., 2006; Holden et al., 2016; Pollard et al., 2013). One-
dimensional energy balance model frameworks have also been used before and are not unique to the CH2O-CHOO TRAIN. Francois and Walker (1992) used an 18-node one-dimensional model coupled to a geochemical model to simulate carbon cycling across the Phanerozoic. This model was subsequently used in other climate (Veizer et al., 2000) and carbon cycle studies, forming the climate component of the COMBINE model (Goddéris and Joachimski, 2004). More recently, Jellinek et al. (2020) used a one-dimensional energy balance model to capture the effect of varying ice cover on climate and weathering.
These one-dimensional frameworks account for spatial dynamics while side-stepping complexity that can obscure cause-and-effect relationships and limit the applications of some higher-dimensional models. However, the water cycle in these previous one-dimensional frameworks was built on approximations largely divorced from physical processes. For example, in Jellinek et al. (2020), precipitation (assumed proportional to runoff) is solved globally and depends only on global mean temperature, whereas in Francois and Walker (1992), runoff depends on an empirical correlation with temperature and latitude
that may not hold in paleoclimate states, particularly under different continental geographies. These model formulations are reasonable solutions to a difficult problem—traditional 1-D energy balance models are known to misrepresent key features of zonal mean hydroclimate (Peterson and Boos, 2020; Siler et al., 2018). Recent energy balance modeling advances, however, address this problem by capturing the spatial complexity of the water cycle in a mechanistic way (Siler et al., 2018). In the CH2O-CHOO TRAIN, we directly employ the one-dimensional energy balance model of Siler et al. (2018), which accurately
simulates meridional atmospheric circulation patterns such as the Hadley cell as well as the spatially distinct precipitation and evaporation responses to warming.

With this improved one-dimensional MEBM, the CH2O-CHOO TRAIN is designed to efficiently explore fundamental interactions between the water cycle, carbon cycle, and climate. The model can simulate about one million years of time in thirty minutes on a standard laptop PC (16 GB RAM, 2.80 GHz processor, without parallelization). Further, its high degree of
customizability makes it well-suited for addressing basic, qualitative questions about the Earth system. Such questions might include the drivers of long-term Cenozoic climate change, the effects of geography on carbon cycling and climate, and the interactions between climatological and geochemical feedbacks. Of course, the model is not optimized for all applications. More specialized and quantitative applications, such as constraining geochemical fluxes from data across a given geologic carbon cycle perturbation event, are limited by the model's customizability because the quantitative results can be sensitive
to somewhat arbitrary initial conditions. In this paper, we outline the model equations and conduct a series of sensitivity tests that explore the features of this coupled climate-carbon cycle system. To emphasize some of the advantages of this model framework, we specifically focus on the effect of climate variables that are often absent from simpler models, and we run simulations spanning about one million years which can be computationally prohibitive in more complex models. We show how continental geography impacts the magnitude and direction of the climate response to changes in certain climate variables,
and how different ice sheet parameterizations affect the response of global temperature, runoff, and the steady state climate to a change in volcanism. Finally, we discuss some of the advantages and limitations of the one-dimensional, zonal mean climate



framework and consider modifications to the climate formulations that can expand the model's potential applications in future work.

## 2 MODEL FORMULATION

The CH2O-CHOO TRAIN links three model frameworks—a model for global climate, weathering, and long-term carbon cycling following Figure 1. The MEBM and weathering models are solved in the zonal mean (1-dimensional; $\sim 200$ km resolution) and integrated to zero dimensions for the global mean box long-term carbon cycle box model (run at 5 kyr timesteps). Geography, climate sensitivity, and other parameters are defined in the MEBM. Geography affects climate via the spatial distribution of albedo and weathering by setting the land area available in a given latitudinal belt. The weathering model receives

inputs of temperature and water runoff from the MEBM and atmospheric $pCO_2$ from the long-term carbon cycle, and outputs fluxes of alkalinity, weathered organic carbon, and phosphorus (P). These fluxes are used calculate the sources and sinks of carbon in the long-term carbon cycle model, which then updates atmospheric $pCO_2$ for use by the MEBM and weathering models. We describe each model in this section with a particular focus on the decisions we make that link the three models together. More detailed descriptions of the individual model frameworks are available from their original publications. Model

code is available on Github and Zenodo (see code availability section and Kukla et al. (2022)), along with instructions for running the model and accessory scripts to generate custom model input files.

### 2.1 Moist Energy Balance Model

#### 2.1.1 Diffusive moist static energy transport

Global climate is simulated in the zonal mean using a Moist Energy Balance Model (MEBM) following the equations and

modifying the code of Roe et al. (2015) and Siler et al. (2018), which built on the earlier work of Flannery (1984) and Hwang and Frierson (2010). Zonal mean atmospheric heating ($Q_{\text{net}}$) is balanced by poleward heat transport on long timescales ($\sim$decadal), yielding equation 1:

$$Q_{\text{net}}(x) = \frac{1}{2\pi a^2} \frac{dF}{dx} \tag{1}$$

where $x$ is the sine of latitude, $a$ is Earth's radius ($m$), $F$ is the column-integrated divergent flux of atmospheric energy

transport ($W$), and $Q_{\text{net}}$ is the difference between top of atmosphere (TOA) and surface net downward energy fluxes ($W\ m^{-2}$) (Pierrehumbert, 2010). When $Q_{\text{net}}$ is positive, atmospheric energy is transported away from $x$, and vice versa.

The MEBM simulates the diffusive transport of the sum of near-surface latent and sensible heat, or moist static energy ($h$; $J\ kg^{-1}$), which is expressed as a function of surface temperature ($T$):

$$h = c_p T + L_v q(T) \tag{2}$$



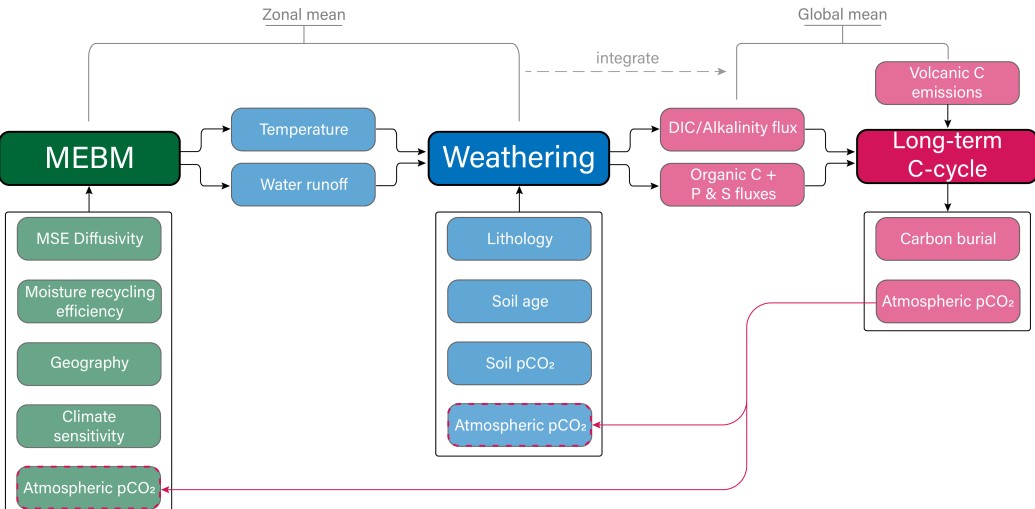

**Figure 1. Coupled model schematic.** The three main model components are labeled "MEBM", "Weathering", and "Long-term C-cycle". Boxes with arrows pointing toward these components include terms required to initialize the model. Arrows pointing out of these components are model output. The MEBM and weathering components are solved in the zonal mean, and then integrated for compatibility with the long-term C-cycle component. Atmospheric $pCO_2$ from the C-cycle component is used as input for the MEBM and weathering components (pink arrows) at the next timestep. MSE is moist static energy; DIC is dissolved inorganic carbon; C and P are carbon and phosphorus, respectively.

where $c_p$ is the specific heat of air ($J\ kg^{-1}$), $L_v$ is the latent heat of vaporization ($J\ kg^{-1}$), and $q$ is the near-surface specific humidity ($g\ kg^{-1}$), calculated as the product of relative humidity and temperature-dependent saturation specific humidity. Relative humidity is used to partition moist static energy into its sensible and latent components and, following previous work, we assume it is constant at the global average, near-surface ocean value of 80% (Hwang and Frierson, 2010; Siler et al., 2018). This fixed humidity assumption is unrealistic over land, particularly in dry regions such as the subtropics, but the assumption

does not interfere with the model's ability to capture the zonal mean aridity profile. Previous work imposed spatially variable relative humidity profiles, (e.g., Peterson and Boos, 2020), but selecting such a profile and how it changes with global climate introduces numerous free parameters. Instead, the Hadley cell parameterization introduced in Siler et al. (2018) (see equation 5 and supplemental text) retains critical features of the zonal mean aridity profile—such as the arid subtropics—and its response to climate while permitting the simplifying assumption of a fixed global humidity value.

Moist static energy (MSE) is transported downgradient (poleward) following:

$$F(x) = -\frac{2\pi p_s}{g} D(1-x^2)\frac{dh}{dx} \tag{3}$$

where $p_s$ is the surface atmospheric pressure ($Pa$), $g$ is gravitational acceleration ($m\ s^{-2}$), and $D$ is a zonally-constant diffusivity coefficient ($m^2\ s^{-1}$).

Equations (1) and (3) can be combined as:





$$Q_{\text{net}}(x) = -\frac{p_s}{ga^2}D\frac{d}{dx}\left[(1-x^2)\frac{dh}{dx}\right].$$ (4)

However, downgradient MSE transport is not valid in the tropics where Hadley circulation promotes upgradient transport of latent heat. With the Hadley cell parameterization of Siler et al. (2018), MSE fluxes are partitioned into a tropical Hadley contribution ($F_{\text{HC}}$) and an extratropical eddy contribution ($F_{\text{eddy}}$) based on a Gaussian weighting function. Net moist static energy transport in the Hadley cell is downgradient, but the latent component of MSE ($F_{\text{HC,q}}$) is transported upgradient following:

$$F_{\text{HC,q}}(x) = -\psi(x)L_v q(x)$$ (5)

where $\psi$ is the southward mass transport in the Hadley cell's lower branch (which equals the northward transport of the upper branch by mass balance). See Siler et al. (2018) and the supplemental text for further details regarding the Hadley cell parameterization.

Finally, by mass balance, the difference between evaporation and precipitation $E - P$ is set equal to the divergence of the
latent component of the MSE flux.

### 2.1.2 Partitioning $P$ and $E$ and parameterizing for land

The divergence of the latent heat flux balances the difference in evaporation and precipitation, or $E - P$, constraining a critical component of the hydrologic cycle that links hydroclimate with the carbon cycle. However, knowledge of $E - P$ is not sufficient to quantify $E$ and $P$ fluxes—doing this requires constraining either $E$ or $P$. We adopt the equation for oceanic evaporation of
Siler et al. (2019) where:

$$E = \frac{R_G\alpha + \rho_{\text{air}}c_p(1-rh)C_H u}{\alpha + \frac{c_p}{L_v/q}}.$$ (6)

Here, $R_G$ is an idealized latitudinal profile of the difference between radiative forcing and the ocean heat uptake response ($W\ m^{-2}$), $\alpha$ is a temperature-dependent Clausius-Clapeyron scaling factor as in Siler et al. (2018, 2019), $\rho_{\text{air}}$ is the near-surface air density ($kg\ m^3$), $rh$ is relative humidity, $C_H$ is a drag coefficient, and $u$ is an idealized surface wind speed profile ($m\ s^{-1}$).
Inputs for the terms defined in equation 6 can be found in the supplementary text.

As discussed earlier, the spatial profile of $E - P$ hinges on assumptions that are based on oceanic conditions (such as constant relative humidity), but it also captures the general trends on land. For example, $E - P$ is generally negative in the tropics and mid-high latitudes and positive in the drier subtropics. However, on decadal timescales on land, $P$ is limited by $E$ such that $E - P$ is always $\leq 0$. In order to calculate terrestrial runoff (an input for the exogenic carbon cycle module) we impose this
mass balance with the Budyko hydrologic balance framework (Broecker, 2010; Budyko, 1974; Fu, 1981; Koster et al., 2006; Roderick et al., 2014; Zhang et al., 2004). We calculate runoff as a fraction of precipitation using the Budyko formulation from Fu (1981):





$$k_{\mathrm{run}} = 1 - \frac{ET}{P} = \frac{E_0}{P} - \left[1 + \left(\frac{E_0}{P}\right)^{\omega}\right]^{1/\omega} - 1. \tag{7}$$

In this formulation, $k_{\mathrm{run}}$ is restricted to $[0,1]$, $ET$ is evapotranspiration, and $E_0$ is potential evaporation. We assume that
ocean evaporation is equal to potential evapotranspiration, setting $E_0$ equal to $E$. The $k_{run}$ term is then used to partition precipitation into runoff and evaporation:

$$q_{\mathrm{land}} = k_{\mathrm{run}} P \tag{8}$$

where $q_{\mathrm{land}}$ is terrestrial runoff, not to be confused with $Q_{\mathrm{net}}$, a radiative forcing term in equation 1. The term $\omega$ in equation 7 is a non-dimensional free parameter with bounds $[1,\infty)$ that represents the proximity of $ET$ to the theoretical limits ($P$ or
$E_0$ when $P < E_0$ and $P > E_0$, respectively). Each value of $\omega$ defines a Budyko curve, with higher values producing a curve where evapotranspiration lies closer to the energy and water limits ($E_0$ and $P$, respectively). We set $\omega$ equal to the global mean value of 2.6 (Budyko, 1974; Zhang et al., 2004; Greve et al., 2015) unless otherwise stated.

### 2.1.3 Greenhouse forcing

Greenhouse gas forcing in our model is driven by the partial pressure of atmospheric $pCO_2$ ($ppmv$). Higher $pCO_2$ decreases
the outgoing longwave flux ($LW_{\mathrm{out}}$; $W\ m^{-2}$) which is assumed to be linearly related to temperature (Budyko, 1969; Koll and Cronin, 2018) by:

$$LW_{\mathrm{out}} = A + BT \tag{9}$$

where $T$ is surface temperature, $B$ is a coefficient that captures the effect of the water vapor feedback, and $A$ is a constant that depends on $CO_2$:

$$A = C_{LW} - M \ln(pCO_{2,t}/pCO_{2,t0}) \tag{10}$$

Here, $C_{LW}$ and $M$ are tunable parameters that determine the climate sensitivity to $pCO_2$. $pCO_{2,t}$ is the partial pressure of $CO_2$ at some time, which is divided by the reference $pCO_2$, $pCO_{2,t0}$. The baseline $pCO_2$ is set at 280 $ppmv$, with parameters $C_{\mathrm{LW}}$, $M$ and $B$ given in supplementary Table S1.

### 2.1.4 Domain boundary conditions

We prescribe the latitudinal distribution of continents and use this distribution to calculate Earth surface albedo. We assign three albedo values—ocean albedo, land albedo, and ice albedo—and calculate the average albedo at each latitudinal node by





the weighted average of land and ocean area. Ice sheets in our model appear at a temperature threshold ($T_{\text{ice}}$, set at $-5°C$) such that the albedo for any node with $T < T_{\text{ice}}$ is equal to the ice albedo (with no dependence on land or ocean values).

The set of MEBM equations can be solved as a boundary value problem, and we use the bvpcol function from the 'bvpSolve'
package in R (Mazzia et al., 2014). We prescribe a zero-flux boundary condition, assuming the flux of moist static energy at both poles is zero. We also prescribe initial temperature guesses for each pole ($T_{\text{north}}$ and $T_{\text{south}}$). The temperature guesses can lead to multi-stability in model solutions. In this paper, we use the same temperature guess for every timestep of a given simulation to enforce a monostable climate (a unique climate solution for every $pCO_2$).

## 2.2 Weathering

We calculate solute concentrations $[C]$ derived from silicate and carbonate weathering at each latitudinal node using each node's surface temperature ($T$) and terrestrial runoff ($q_{\text{land}}$). $T$ and $q_{\text{land}}$ are derived from the MEBM (see Section 2.1.2).

The value of $[C]_{\text{sil}}$ is calculated using equations modified from Maher and Chamberlain (2014) (similar to the "MAC" model in other works (Baum et al., 2022; Graham and Pierrehumbert, 2020)). These equations permit us to explicitly incorporate the effect of $T$, $q_{\text{land}}$, and weathering zone $pCO2$—variables that are all influenced by atmospheric $pCO_2$ and climate—on $[C]_{\text{sil}}$:

$$[C]_{\text{sil}} = [C]_{\text{sil,eq}} \left( \frac{\frac{Dw}{q}}{1 + \frac{Dw}{q}} \right) \tag{11}$$

Here, $[C]_{\text{sil,eq}}$ is the maximum, equilibrium concentration of silicate-derived bicarbonate (Maher, 2011) and $Dw$ is the Damköhler weathering coefficient, which is a term that encapsulates the reactivity of the weathering zone and the time required to reach equilibrium. Following Maher and Chamberlain (2014), we define $Dw$ as:

$$Dw = \frac{L_\phi r_{\text{max}} \frac{1}{1 + t_{\text{wz}} m k_{\text{eff}} A}}{[C]_{\text{sil, eq}}} \tag{12}$$

where $L\phi$ is the reactive length scale, $r_{\text{max}}$ is the theoretical maximum reaction rate, $t_{\text{wz}}$ is the age of the weathering zone and is a key variable describing the reactivity of the weathering zone, $m$ is the molar mass of weathering minerals, $A$ is the specific surface area of minerals undergoing weathering, and $k_{\text{eff}}$ is the effective reaction rate. In this model, $L\phi$ and $r_{\text{max}}$ are held constant. This effective reaction rate is defined as an Arrhenius function that describes the temperature dependency of reaction rates (Brady, 1991; Kump et al., 2000):

$$r_{\text{eff}} = k_{\text{reac}} e^{\left[ \left( \frac{Ea}{R_g} \right) \left( \frac{1}{T_0} - \frac{1}{T} \right) \right]} \tag{13}$$

where $R_g$ is the universal gas constant ($J\ K^{-1}\ mol^{-1}$) (distinct from $R_G$ in equation 6) and $Ea$ is the activation energy ($J\ mol^{-1}$). The coefficient $k_{\text{reac}}$ ($yr^{-1}$) encapsulates the effects of mineral surface area, molar mass, and the reference reaction rate (all assumed constant) in modulating the effect of temperature on reaction rate.



Lastly, $[C]_{\text{sil,eq}}$ is modified by the availability of reactant, which here is assumed to be primarily $CO_2$. We calculate this effect as a function of weathering zone $pCO_2$ assuming open-system $CO_2$ dynamics, following Winnick and Maher (2018):

$$[C]_{\text{sil,eq}} = [C]_{\text{sil,eq,0}} \left( R_{\text{CO2, wz}} \right)^{0.316} \tag{14}$$

where $[C]_{\text{sil,eq,0}}$ is the pre-perturbation, initial value of $[C]_{\text{sil,eq}}$. $R_{\text{CO2,wz}}$ is the ratio of weathering zone $pCO_2$ at time t ($WZ_{\text{CO2}}$) to the initial weathering zone $pCO_2$ pre-perturbation ($WZ_{\text{CO2,0}}$). The exponent value of 0.316 is derived by Winnick and Maher (2018) based on the net weathering stoichiometry for an open-system scaling relationship for the dissolution of plagioclase feldspar ($An_20$) and precipitation of halloysite. Depending on the primary lithology and secondary mineral precipitated this exponent can vary from $\sim 0.25$ to $0.7$ (see Winnick and Maher (2018); their Table 1). This average stoichiometry represents an average granodiorite continental crust (e.g., Maher, 2010, 2011; Maher and Chamberlain, 2014). We calculate $R_{\text{CO2,wz}}$ using a formulation proposed by Volk (1987) that links weathering zone $pCO_2$ with the primary source of that $CO_2$, which is aboveground terrestrial gross primary productivity ($GPP$). Here, $WZ_{\text{CO2}}$ is calculated using an equation that links $GPP$, $CO_2$ fertilization of $GPP$, and weathering zone $CO_2$:

$$WZ_{\text{CO2}} = \left[ R_{\text{GPP}} \left( 1 - \frac{pCO_{2,0}}{WZ_{\text{CO2,0}}} \right) + \frac{pCO_2}{WZ_{\text{CO2,0}}} \right] WZ_{\text{CO2,0}} + \left( pCO_2 - pCO_{2,0} \right) \tag{15}$$

Here, $R_{\text{GPP}}$ is the ratio of $GPP$ at time $t$ to the pre-perturbation $GPP$ ($GPP_0$) and the last term on the right-hand side of the equation ensures that $WZ_{CO2}$ is always greater than atmospheric $pCO_2$. The $GPP$ is calculated using a Michaelis-Menton formulation:

$$GPP = GPP_{\text{max}} \left[ \frac{pCO_2 - pCO_{2,\text{min}}}{pCO_{2,\text{half}} + \left( pCO_2 - pCO_{2,\text{min}} \right)} \right] \tag{16}$$

where $GPP_{max}$ is the maximum possible global terrestrial $GPP$, $pCO_{2,min}$ is the $pCO_2$ at which photosynthesis is balanced exactly by photorespiration, and $pCO_{2,half}$ is the $pCO_2$ at which $GPP$ is equivalent to 50% $GPP_{max}$:

$$pCO_{2,\text{half}} = \left( \frac{GPP_{\text{max}}}{GPP_0} - 1 \right) \left( pCO_{2,0} - pCO_{2,\text{min}} \right) \tag{17}$$

We choose a $pCO_{2,\text{min}}$ of 100 ppm based upon evidence for widespread $CO_2$ starvation at the Last Glacial Maximum (LGM) (Prentice and Harrison, 2009; Scheff et al., 2017), which had an atmospheric $pCO_2$ of 180 ppm. We also assume that $GPP_{\text{max}}$ is equal to twice $GPP_0$, though our results are insensitive to this parameter. Lastly, we assume that $WZ_{\text{CO2,0}}$ is a factor of 10 larger than atmospheric $pCO_{2,0}$ given evidence that soil $pCO_2$ is typically elevated above atmospheric levels by approximately an order of magnitude (Brook et al., 1983).

We use this set of equations to calculate silicate and carbonate weathering. We parameterize maximum carbonate weathering reaction rates as being 1000 times faster than silicate weathering (Lasaga, 1984; Morse and Arvidson, 2002), carbonate $[C]_{\text{sil,eq}}$





approximately 2 times greater than for silicate weathering (Ibarra et al., 2017), and a reactive length scale, $L\phi$, 10 times greater than for silicates. The concentrations, $[C]$, calculated above are translated into global weathering fluxes ($F_w$):

$$F_{\text{w,sil/carb}} = q_{\text{land}}(x)C_{\text{sil/carb}}(x)A_{\text{land}}(x)W_{\text{sil/carb}} \tag{18}$$

where $x$ is the sine of latitude following the MEBM grid spacing, subscripts $sil$ and $carb$ refer to silicate and carbonate
weathering, respectively, $A_{\text{land}}$ is the land area, and $W$ is a scalar used to enforce mass balance. The $W$ parameter is a global constant and differs for carbonate vs silicate weathering. We calculate $W_{\text{sil/carb}}$ during model initialization to ensure that the global sum of carbon burial fluxes equals the sum of input fluxes, such that the model starts in steady state. For silicate weathering, $W_{\text{sil}}$ scales with $F_{\text{wsil}}$ such that at initialization, $F_{\text{wsil}}$ equals $F_{\text{volc}}$. The $W_{\text{sil}}$ scalar can also be thought of as loosely representing a global $SiO_2{:}HCO_3$ ratio that translates silica fluxes to carbon fluxes. This translation is necessary because these
weathering equations—and the associated parameters—in Maher and Chamberlain (2014) were originally derived for Si fluxes, rather than C (or alkalinity) fluxes. Ibarra et al. (2016) demonstrated that for modern basaltic and granitic catchments $[C]_{\text{sil,eq,0}}$ scales proportional to weathering stoichiometry, as predicted by Winnick and Maher (2018), and Dw scales with some bias towards more chemostatic (higher $Dw$ values) in Si compared to alkalinity (Moon et al., 2014). Because $W_{\text{sil}}$ determines the sensitivity of weathering fluxes to changes in runoff and concentration, it also influences the strength of the silicate weathering
feedback (defined as the change in weathering fluxes per change in atmospheric $CO_2$). Similarly, $W_{\text{carb}}$ is determined by scaling the sum of the carbonate weathering fluxes at each node such that these fluxes equal the estimate of the carbonate weathering flux in Wallmann (2001), forcing the model to start in steady state.

## 2.3 Carbon cycle

The carbon cycle model follows other one-box models that are commonly employed for tracking long-term (*i.e.*, on timescales
of $> 10^5$ years) changes to the carbon cycle and $\delta^{13}C$ (e.g. Berner, 1991; Kump and Arthur, 1999). The input fluxes of C into the ocean-atmosphere system include volcanism and solid Earth degassing ($F_{\text{volc}}$), organic carbon weathering ($F_{\text{w,org}}$), and carbonate weathering ($F_{\text{w,carb}}$), and the output fluxes are the burial of organic carbon and carbonate carbon in marine sediments ($F_{\text{b,org}}$ and $F_{\text{b,carb}}$, respectively). The input fluxes have an associated $\delta^{13}C$ (*i.e.*, $\delta^{13}C_{\text{volc}}$, $\delta^{13}C_{\text{w,org}}$ and $\delta^{13}C_{\text{w,carb}}$) and the $\delta^{13}C$ of the output fluxes are determined by a fixed fractionation factor relative to the global average of the ocean-atmosphere
system ($\epsilon = \delta^{13}C - \delta^{13}C_{\text{output flux}}$ *i.e.*, $\epsilon_{\text{b,org}}$ and $\epsilon_{\text{b,carb}}$). The subsequent mass balance equation for the total mass of carbon in the one-box ocean-atmosphere ($M_C$) is

$$\frac{dM_C}{dt} = F_{volc} + F_{worg} + F_{wcarb} - F_{borg} - F_{bcarb}. \tag{19}$$

The associated isotope mass balance equation for the carbon isotope value of the ocean-atmosphere system ($\delta^{13}C$) is

$$\frac{d\delta^{13}C}{dt}M_C = F_{\text{volc}}(\delta^{13}C - \delta^{13}C_{\text{volc}}) + F_{\text{w,org}}(\delta^{13}C - \delta^{13}C_{\text{w,org}}) + F_{\text{w,carb}}(\delta^{13}C - \delta^{13}C_{\text{w,carb}}) - F_{\text{b,org}}\epsilon_{\text{b,org}} - F_{\text{b,carb}}\epsilon_{\text{b,carb}}. \tag{20}$$





For simplicity the organic carbon burial flux ($F_{\text{b,org}}$) is scaled to the carbonate burial flux ($F_{\text{b,carb}}$) (Caves Rugenstein et al., 2019; Ridgwell, 2003). The carbonate carbon burial flux is a function of the calcite saturation index, ($\Omega_{\text{calcite}}$), such that $F_{\text{b,carb,t}} = F_{\text{b,carb,i}} \times \Omega_{\text{t}}/\Omega_{\text{i}}$ where subscripts t and i refer to some point in time and the initial condition, respectively. The $\Omega_{\text{calcite}}$ is calculated as a function of the carbonate system (Zeebe and Wolf-Gladrow 2001), and we correct for the concentrations of [Mg2+] and [Ca2+] using the equations of Zeebe and Tyrrell (2019). The alkalinity reservoir in the ocean, $M_{\text{Alk}}$ is related to

the fluxes by:

$$\frac{dM_{\text{Alk}}}{dt} = F_{\text{w,sil}} + F_{\text{w,carb}} - F_{\text{b,carb}}, \tag{21}$$

where $F_{\text{w,sil}}$ is the silicate weathering flux. Parameter values and references can be found in Supplementary Table S3 and specific details about key parameters are described below.

To achieve mass balance for $M_C$ and $M_{\text{Alk}}$, $F_{\text{volc}}$ must equal $F_{\text{w,sil}}$ at steady state. The global temperature at Earth's surface

($T_a$), is calculated by integrating the MEBM temperature results weighted by land area, and is then used to set the global mean ocean temperature $T_o$ by assuming the mean ocean temperature is $10°C$ colder than $T_a$. To solve the initial carbonate system and associated initial $M_C$ and $M_{Alk}$, the initial ocean pH, $pCO_2$, $T_o$, salinity (35 p.s.u.), mean ocean pressure (300 bar), and geochemical composition of seawater (*i.e.*, Ca = 15 moles/L, Mg = 48.5 moles/L, and $SO_4^{2-}$ = 28.2 moles/L) are calculated using the speciation equations of Zeebe and Wolf-Gladrow (2001), modified by Zeebe and Tyrrell (2019) to account

for variable ocean chemistry (see Supplementary Table S3).

## 2.4    Coupled climate-carbon cycle model initialization and integration

We first initialize the MEBM and carbon cycle boundary conditions including the initial global carbon cycle fluxes (see Supplementary Tables S3 and S1). The carbon cycle is parameterized to start in steady state (inputs of carbon = outputs). We begin by simulating the initial climate state by prescribing an initial atmospheric $pCO_2$ to force the MEBM. This same $pCO_2$ is used,

along with an initial pH (Supplemental Table S3), to speciate the carbon cycle (Zeebe and Wolf-Gladrow, 2001). Temperature and runoff from this initial climate state are used to calculate weathering fluxes following equation 18 where the scalar $W$ is set to one. We then calculate the scalars for carbonate and silicate weathering by:

$$W_{\text{sil}} = \frac{F_{\text{volc, ss}}}{\sum F_{\text{w,sil,i}}} \tag{22}$$

$$W_{\text{carb}} = \frac{F_{\text{w,carb, ss}}}{\sum F_{\text{w,carb,i}}} \tag{23}$$

where $i$ refers to the initial, unscaled fluxes, the $ss$ subscript refers to the steady state fluxes, and the $\sum$ denotes the global sum (note that the zonal mean grid is an equal-area grid).





We couple the MEBM to the long-term carbon cycle module by solving the MEBM at each timestep within the carbon cycle solver. First, the carbon cycle module solves for atmospheric $pCO_2$ for the given timestep using the mass balance of DIC and alkalinity and the carbonate speciation described above. This $pCO_2$ value is used to force the MEBM. Based on the MEBM results, the polar temperature guesses ($T_{north}$ and $T_{south}$) are either updated (if the ice configuration has changed) or remain unchanged (if the ice configuration is the same). In this case, the ice configuration determines the state of the global climate system. Some temperature guesses lead to no stable solution or a "snowball Earth" configuration where the entire planet is glaciated (Supplemental text). For either of these outcomes, we re-run the MEBM using temperature guesses that are a small step (usually $\sim 0.5°C$) toward a warmer direction. This is repeated until the MEBM finds a stable solution that is not a fully-glaciated planet (usually less than 3 steps are needed until such a solution is reached). We use this method for avoiding snowball states because the range of $pCO_2$ forcing in our simulations is above the lowest $pCO_2$ concentrations of the Quaternary where a fully-glaciated planet is unreasonable, although users may easily turn off this snowball-avoiding feature. The fact that fully-glaciated solutions are usually not robust to small perturbations in the temperature guess suggests that our simulations do not approach a true snowball scenario. Once a stable solution is found, the latitudinally-resolved hydrological and temperature output from the MEBM are used to solve for the silicate and carbonate weathering fluxes. These weathering fluxes, plus any perturbations to the input carbon fluxes (such as via changes in $F_{volc}$) then drive the response of the long-term carbon cycle.

## 2.5 Model assumptions and limitations

A number of model processes are not fully coupled among all modules. These processes are parameterized with simplifying assumptions for the purpose of this work, but could be coupled in the future (albeit with additional parameters). We detail these assumptions below.

We assume that no weathering occurs beneath ice sheets such that the weathering fluxes at glaciated latitudes are set to zero. While weathering rates in glacial catchments can be high, it remains unclear whether glaciated catchments are a net source or sink of $CO_2$ on long timescales (Torres et al., 2017). Our assumption of no weathering beneath ice sheets is consistent with previous modeling work (e.g., Zachos and Kump, 2005; Pollard et al., 2013) and it has two main effects on our model. First, when a simulation transitions from a greenhouse to an icehouse there is a reduction in weathering due to cooler climate conditions and decreasing precipitation *and* an additional reduction due to ice sheet growth. Second, when initializing the model in an icehouse state, the weathering scalar term, $W$ (calculated at initialization), is higher because the denominator in equations 22 and 23 is lower due to ice coverage. We test the sensitivity of this assumption to our results in our model experiments (next section).

We also neglect the effect of ice coverage on global eustatic sea level in our model. Terrestrial ice coverage decreases sea level, exposing more land area and potentially increasing global weathering. This effect would counteract the effect of ice coverage decreasing the weatherable land area, discussed above. However, accounting for the effect of sea level on exposed land area requires constraints on hypsometry (at least near the coasts) as well as ice volume, both of which are absent from our 1-dimensional model framework. Moreover, any increase in silicate weathering due to sea level fall is expected to be small or



negligible because exposed shelves are likely to consist of carbonates and organic and clay-rich sediment, not primary silicate minerals (Berner, 1994; Gibbs and Kump, 1994; Kump and Alley, 1994), and may act as a source of atmospheric $CO_2$ (Kölling et al., 2019).

Hydrological fluxes, land albedo, and GPP are not coupled in our model and precipitation does not affect land albedo or GPP.
A wetter climate is expected to decrease land surface albedo by supporting greater leaf area and therefore a darker surface, whereas a drier climate tends to have the opposite effect (e.g. Charney, 1975; Claussen, 1997). Decoupling these processes in our model means that precipitation is not responsive to vegetation (*i.e.* no precipitation-vegetation albedo feedback) and the weathering response to hydrological fluxes is solely due to their effect on runoff, with no indirect additional effect via GPP (see equation 16) and soil $pCO_2$. Including such a parameterization would likely heighten the sensitivity of weathering to
hydrologic change.

Part of the weathering module scales silicate and carbonate solute concentrations with weathering zone $pCO_2$. This weathering zone $pCO_2$ is calculated from several global parameters, including atmospheric $pCO_2$ and $GPP$, and ignores local climatic influences on weathering zone $pCO_2$. Soil $pCO_2$ is known to vary with local climate (Brook et al., 1983; Cotton and Sheldon, 2012; Cotton et al., 2013) and decoupling weathering zone $pCO_2$ from local climate is clearly a major simplification.
Nevertheless, there remains substantial uncertainty regarding how soil zone $pCO_2$ will change in response to warming and rising atmospheric $pCO_2$ (Terrer et al., 2021). This uncertainty motivates our use of the simpler, global model for GPP following (Volk, 1987, 1989). Given that $[C]$ is, in our model, only sensitive to weathering zone $pCO_2$ to the power of 0.316 (Winnick and Maher, 2018), the lack of a coupling between local climate and weathering zone $pCO_2$ is likely to have a muted effect on our predicted $[C]_{\text{sil}}$ and $[C]_{\text{carb}}$.

An additional assumption in our model is that the negative feedback that regulates long-term climate is terrestrial weathering (*i.e.* continental silicate and carbonate weathering). For example, we have not explicitly included seafloor basalt weathering fluxes as a silicate weathering flux separate from terrestrial silicate weathering fluxes, though recent work suggests that seafloor basalt weathering may be a substantial portion of the global weathering flux, particularly during hothouse climates (Coogan and Gillis, 2013, 2018). Further, we assume that the positive and negative feedbacks of the organic carbon cycle, other than
burial, are outpaced by the silicate weathering feedback on climate. Recent work has suggested that organic carbon weathering may be linked to climate (Hilton and West, 2020) and that terrestrial organic carbon export to continental shelves is partly influenced by climate (Galy et al., 2015). This coupling between climate and organic carbon weathering—as well as links to marine productivity and organic carbon export and burial in marine sediments—remains an area of intensive research, and introducing carbon cycle feedbacks on climate via the organic carbon cycle represents a promising avenue for further work. We
do note that, in our model, there is a simplified coupling between climate and organic carbon burial ($F_{\text{b,org}}$) as $F_{\text{b,org}}$ is linked to $F_{\text{b,carb}}$, which is sensitive to $\Omega$ and, hence, to atmospheric $pCO_2$ and to $F_{\text{w,carb}}$, whereas other frameworks simulate organic carbon burial more mechanistically, explicitly capturing features absent from our model such as diagenesis, redox-dependence of Phosphorus cycling, and more (Hülse et al., 2018).



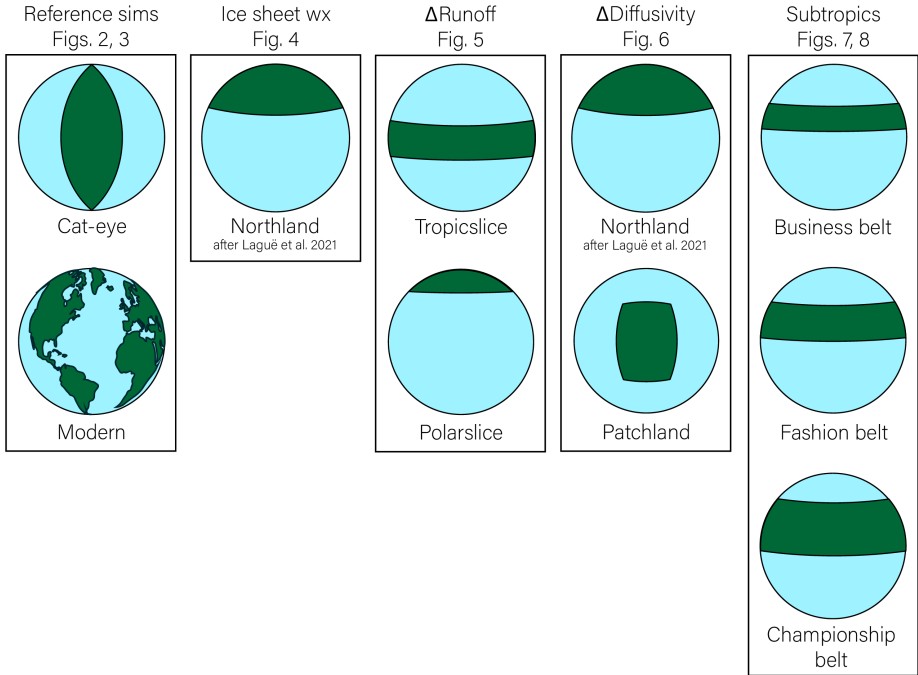

**Figure 2. List of continental configurations**. Each box refers to a set of model experiments, labelled with the associated figures. Maps are shown in two-dimensions for convenience, but are simulated in the one-dimensional, zonal mean framework.

## 3   MODEL EXPERIMENTS

We run a series of experiments with the CH2O-CHOO TRAIN model to demonstrate its features and sensitivity to key climate variables. We use different continental configurations for each set of experiments to emphasize how spatially variable temperature and runoff responses can impact weathering fluxes (Fig. 2). These experiments are not meant to be an exhaustive sensitivity analysis of the model. Instead, they provide a baseline for model performance and they illustrate climate-carbon cycle interactions that are absent in many simpler models and emergent properties of more complex frameworks.

In our first set of experiments, our reference simulations, we begin by analyzing the spatial pattern of climate and weathering for a "Cat-eye" geography at different glacial conditions and $pCO_2$ levels. In the Cat-eye geography, each latitudinal band has the same proportion of land and ocean and, therefore, also has the same average albedo. This idealized geography allows us to explore the basic spatial pattern of temperature, hydroclimate (precipitation and evaporation), and silicate weathering and their response to greenhouse forcing without introducing additional spatial complexity from the distribution of land. Next, we

test the model using modern geography and imposing a perturbation similar to the Paleocene-Eocene Thermal Maximum with an injection of 5000 Pg of carbon to the atmosphere over 10000 years. This simulation is used as a verification of the coupled model's performance in comparison with other, similar simulations across the model hierarchy.





Second, we evaluate the sensitivity of our model results to the effect of ice sheets on weathering. In these experiments, we use a "Northland" geography, inspired by Laguë et al. (2021), where land covers all area from the north pole down to 380  12°N. This geography is useful for testing the effect of continental ice sheets on weathering because much of the land area is concentrated in the high latitudes where ice sheets can modify local weathering fluxes. For these simulations, we force climate with an instantaneous, permanent halving of the volcanic flux and we vary how much ice sheets decrease the runoff available for chemical weathering, or effective runoff. If ice sheets fully inhibit weathering, then the effective runoff passed to the weathering model is zero for the latitudes where ice cover exists. In contrast, if runoff from ice sheets drives weathering 385  fluxes similar to the ice-free scenario, then the effective runoff passed to the weathering model is the same as the MEBM output. A higher effective runoff value at ice-covered latitudes can also be thought of as due to ice sheets decreasing sea level to expose weatherable continental shelf, or only partial ice cover over land.

Third, we test the model response to an instantaneous change in runoff by modifying the efficiency of moisture recycling ($\omega$ in equation 7). The term $\omega$ determines how efficiently precipitation is partitioned into evaporation versus runoff, with higher values of $\omega$ corresponding with more evaporation and less runoff. For this experiment, we test two different geographies characterized by a thin, 20°latitude slice of land either in the tropics ("Tropicslice") or north pole ("Polarslice"). We use these geographies because the runoff and temperature response to climate is different in the tropics versus the poles, so these configurations are end-member cases for exploring how temperature, runoff, climate, and geography interact. In each geographic configuration, we change $\omega$ from the control value of 2.6 to 2.0 (increasing runoff) and to 3.5 (decreasing runoff), 395  approximating the 25th and 75th quantiles of the global distribution of $\omega$ (Greve et al., 2015).

Fourth, we test the model response to an instantaneous change in the diffusivity of moist static energy ($D$ in equation 3). An increase in diffusivity tends to cool the tropics, warm the poles, and transport more moisture from the subtropics to the mid-to-high latitudes. To capture these spatially complex effects, we simulate a change in $D$ for two geographies—a northern hemisphere continent (Northland, as in the ice sheet simulations) and a tropical + subtropical patch of land ("Patchland"). 400  Patchland does not extend across the whole planet so as to keep the global land area more similar between the two geographies. Unlike the Tropicslice and Polarslice geographies, Patchland and Northland each span more than one climate zone. In these experiments, we vary $D$ by $\sim \pm 30\%$ from the control value of $1.06 \times 10^6 \ m^2 \ s^{-1}$ (Hwang and Frierson, 2010) consistent with variations one might expect between an icehouse and a greenhouse climate (e.g., Frierson et al., 2006).

Finally, we explore a case study where the negative silicate weathering feedback on climate breaks down. Silicate weathering 405  is often assumed to increase with $pCO_2$ and/or temperature in simpler models (Berner, 2006; Bergman, 2004; Caves Rugenstein et al., 2019), but some work has argued for cases where weathering can decrease with warming, causing a runaway positive feedback (Kump, 2018; Mills et al., 2021; Pollard et al., 2013). Such a scenario can be simulated in the CH2O-CHOO TRAIN when most land available for weathering is situated in the subtropics, where the model predicts that runoff typically decreases with warming. We repeat the instantaneous change in $\omega$ simulations from above using smaller changes in $\omega$ (from 410  $\omega = 2.6$ to 2.4 and 3.0) and with three geographies with belts of land of differing widths, all centered on the northern hemisphere subtropics. We test a smaller change in $\omega$ than above because the weathering feedback is weak in these simulations due to the geography, so small perturbations cause larger changes in climate. The three geographies include a narrow belt (Business





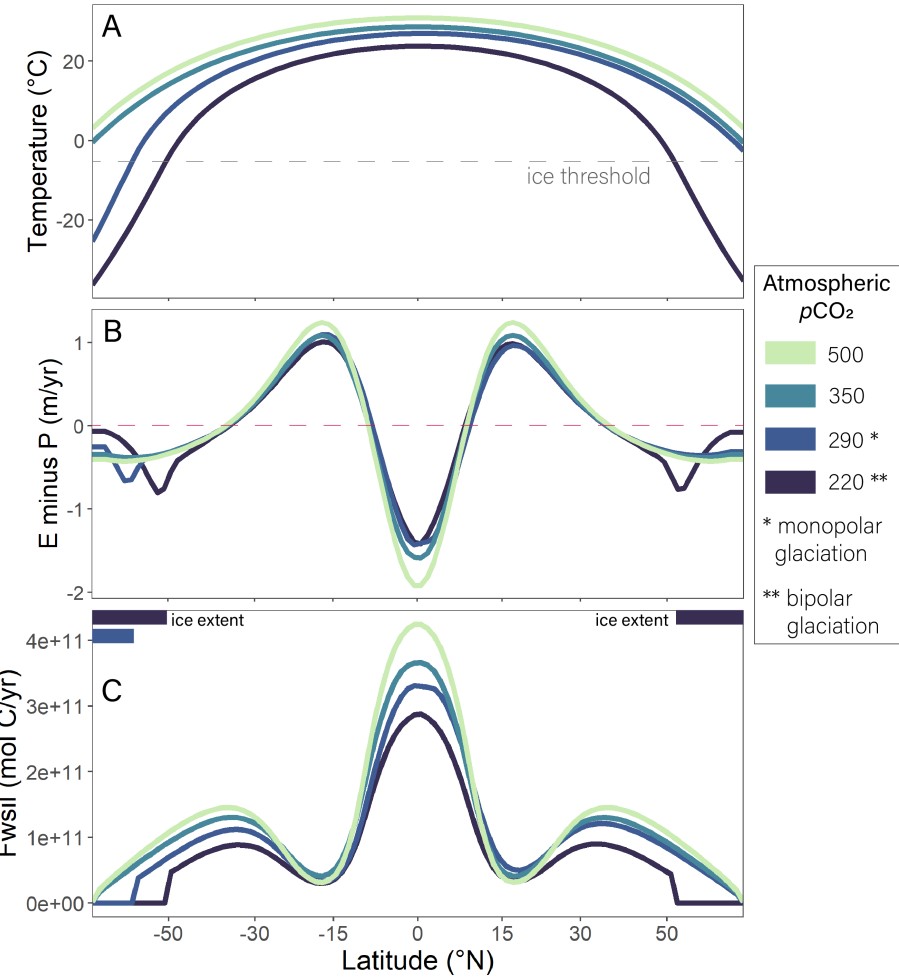

**Figure 3. Reference climate state**. Zonal mean output for (**A**) temperature, (**B**) evaporation minus precipitation, and (**C**) the silicate weathering flux are shown for four $pCO_2$ levels. Negative values of E minus P indicate more runoff. The E minus P values shown here are output from the MEBM and do not include the water availability limit of the Budyko hydrologic balance framework applied over continents.

belt world) spanning 10-30°N, a mid-width melt (Fashion belt world) spanning 5-35°N, and a broad belt (Championship belt world) spanning the equator to 40°N.

**4  RESULTS**

**4.1  Reference Simulation**

We define a reference simulation with the cat-eye geography to illustrate the one-dimensional outputs of the MEBM and weathering models (Fig. 3). Global climate transitions from a bipolar glaciation (similar to present-day) to a monopolar glaciation



and, finally, ice-free as $pCO_2$ increases from 220 to 350 ppmv. The $pCO_2$ thresholds for glaciation are strongly dependent
on the prescribed climate sensitivity as well as the geography and prescribed land, ocean, and ice albedo values, among other
factors. Despite the land cover and insolation forcing being meridionally symmetric about the equator, asymmetric results such
as the monopolar glaciation are still possible. The monopolar state exists because there is too much atmospheric $pCO_2$ to sup-
port a colder, bipolar glaciation, but too little to support an ice-free world. When the land cover (thus, albedo) and insolation
are meridionally symmetric, the pole with the lower initial temperature guess will glaciate first as the planet cools. If both
temperature guesses are the same, the first pole to glaciate in the meridionally symmetric case will depend on the tuning of
the numerical solver. However, the vast majority of model cases likely involve meridional asymmetry, as continental geogra-
phies through time are never perfectly symmetric about the equator. In such asymmetric cases, the pole that glaciates in the
monopolar case is determined by the asymmetry, not the polar temperature guess and numerical tuning.

The model hydroclimate output is shown in the spatial pattern of $E$ minus $P$, which balances the divergence of the latent
heat flux (Fig. 3B). $P$ exceeds $E$ in the tropics and mid-to-high latitudes, but $E$ is greater than $P$ in the subtropics due to the
dry downwelling branches of the Hadley cell. There is an abrupt decrease in $E$ minus $P$ at the ice threshold due to the step-wise
change in albedo, temperature, and moist static energy which forces rainout. The global temperature and hydroclimate fields
shown in Fig. 3A and B ultimately determine the spatial pattern of silicate weathering (Fig. 3C). Weathering fluxes are highest
in the tropics where temperature and runoff are high, and lowest at the poles where temperature and runoff are low. Whereas
the broad spatial pattern of runoff sets the pattern of silicate weathering, changes in silicate weathering with climate largely
respond to temperature in these simulations because the runoff response is small. For example, runoff is generally insensitive
to global climate between 30 and 50 degrees latitude (north or south) (Fig. 3B), but weathering fluxes increase with $pCO_2$
due to the combined effect of warmer temperatures and higher soil $pCO_2$ (Fig. 3C). Weathering fluxes are zero for glaciated
latitudes because, for these simulations, we assume zero effective runoff for ice-covered latitudes.

**4.2 Response to abrupt $pCO_2$ increase with modern geography**

Starting at 320 ppmv $pCO_2$ for the modern geography, the model simulates a bipolar glaciation with the spatial pattern of
global discharge and silicate weathering closely matching that of continental area and the zonal mean water balance (Fig. 4A-
C). Today, mean air temperatures in the south pole are lower than in the north pole, whereas the model finds a cooler north pole.
This discrepancy is probably due to the fact that we do not account for factors such as ocean circulation, spatial variability in
land albedo, cloud feedbacks, or cloud albedo. For example, in the model more land in the northern hemisphere leads to higher
albedo and a cooler climate, although recent work shows that more cloud cover in the southern hemisphere compensates for
the effect of northern hemisphere land, causing both hemispheres to have approximately the same top of atmosphere albedo
(Datseris and Stevens, 2021).

When forced with an injection of carbon similar in magnitude to the Paleocene Eocene Thermal Maximum (PETM; ~5 Pg
over 10 kyr), global climate and the carbon isotope composition of the DIC pool recover in approximately 200-300 kyr (Fig.
4D, G). During this time, ice sheets fully melt as the planet warms to a greenhouse climate and then are reestablished, first in
the northern hemisphere at ~1500 ppmv $pCO_2$ and later in the southern hemisphere. We note that which hemisphere glaciates



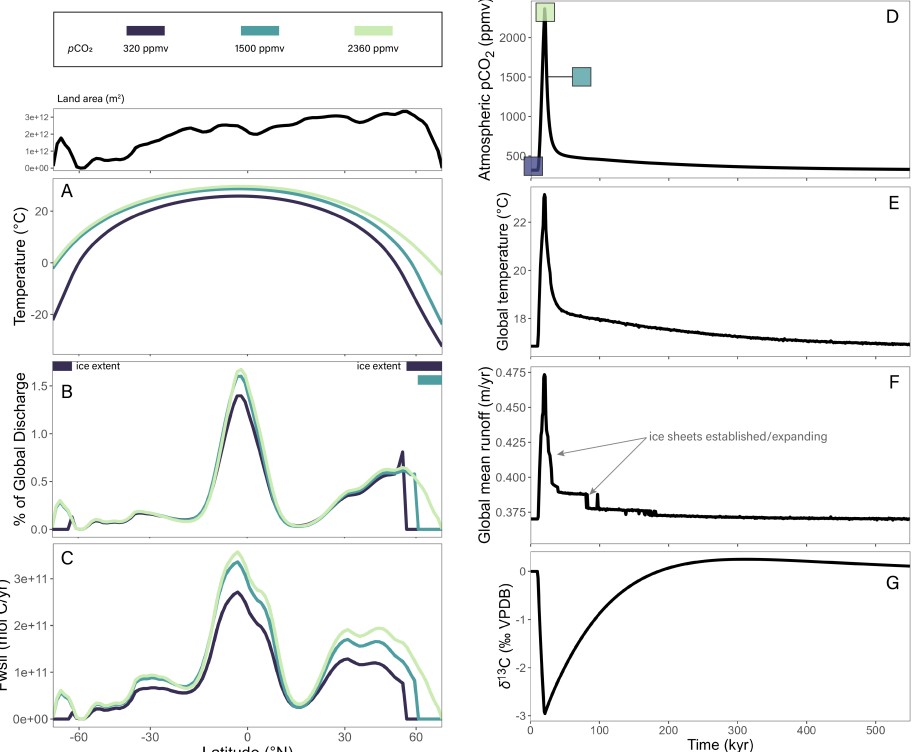

**Figure 4. Abrupt warming experiment for the modern world**. Zonal mean results for **(A)** temperature, **(B)** the percent of global discharge at each latitudinal grid cell, and **(C)** the silicate weathering flux at three selected timesteps. The timesteps in *(A-C)* are labeled with colored boxes in panel **(D)** which shows the time evolution of atmospheric $pCO_2$, along with that of global temperature in **(E)**, global mean runoff in **(F)**, and the carbon isotope composition of the DIC pool. Arrows denote stepwise changes in runoff due to the establishment of ice sheets limiting runoff beneath them.

first is not sensitive to the initial temperature conditions, as in the meridionally symmetrical geography case (Fig. 3). The $pCO_2$ thresholds for glaciation are also sensitive to model tuning, particularly due to changes in the prescribed climate sensitivity. A

higher climate sensitivity will decrease the $pCO_2$ level at which ice melts as well as the maximum $pCO_2$ reached during the same perturbation.

## 4.3 Varying the effect of ice cover on weathering

When we halve the volcanic flux of $CO_2$ in our model, allowing ice cover to increase, the temperature and $pCO_2$ of the new equilibrium climate state depends on how much ice cover decreases the runoff available to weather rock, or "effective

runoff". As the effective runoff (and therefore weathering) decreases at ice covered latitudes, the new equilibrium climate state is warmer with less ice and higher $pCO_2$ than if the effective runoff at these latitudes is higher (Fig. 5). The instantaneous decrease in the volcanic flux leads to lower $pCO_2$ and global cooling which continues until silicate weathering fluxes decrease





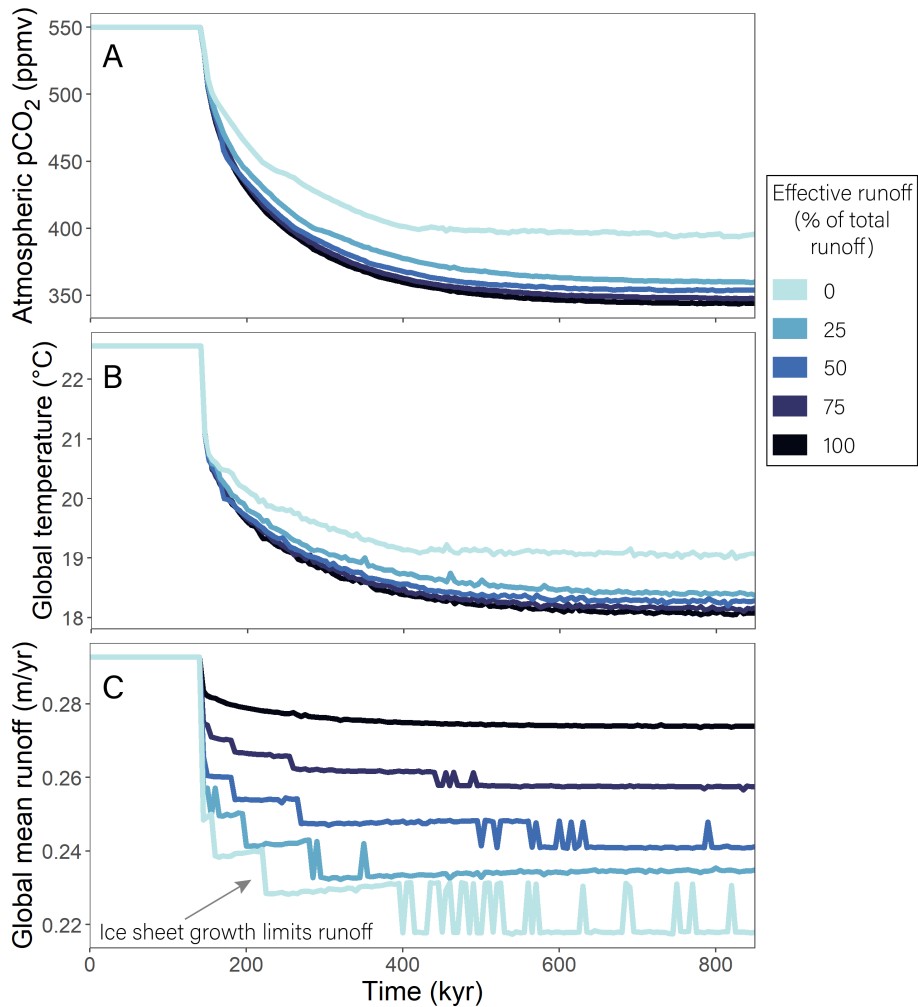

**Figure 5. Effect of ice sheets inhibiting runoff**. Time-evolution of atmospheric $pCO_2$ **(A)**, global mean temperature **(B)**, and global mean runoff **(C)** for different effective runoff scenarios when the volcanic input is halved. When effective runoff at ice-covered latitudes is zero percent of total runoff (light blue line) a smaller change in temperature is needed to balance the carbon cycle. The largest change is necessary when effective runoff is equal to the total runoff predicted by the MEBM (black line).



sufficiently for carbon burial to balance the new, lower volcanic emissions flux. When there is zero effective runoff due to ice cover (ice sheet fully inhibits weathering across the latitudinal band; light blue line of Fig. 5), ice sheet growth causes a greater

decrease in weathering for the same decrease in temperature. Consequently, the equilibrium weathering flux corresponds to a higher global temperature than if the ice sheet has a smaller inhibition on runoff. Thus, we find that ice sheets that fully inhibit effective runoff lead to the largest change in runoff yet the smallest change in temperature.

These simulations are run with the Northland continental geography to maximize the effect of polar ice sheets on runoff. As a result, the growth of the ice sheet causes stepwise decreases in the global runoff response, with larger steps occurring

when the ice sheet more strongly inhibits runoff (Fig. 5C). Ice sheet growth is stepwise in the model because of the positive ice albedo feedback whereby ice expansion causes cooling which causes further ice expansion. Changes in effective runoff would cause no difference in the new equilibrium climate state if we repeated these simulations for a continental geography such as Tropicslice world where the land remains ice-free at these levels of volcanic forcing.

### 4.4 Instantaneous change in moisture recycling efficiency

Figure 6 shows the model climate and carbon response to an abrupt increase (panels A-D) and decrease (panels E-H) in runoff, as determined by the recycling efficiency parameter $\omega$. When $\omega$ decreases from 2.6 to 2.0, runoff increases everywhere driving more weathering. However, the magnitude of the weathering (and thus climate) response to changing $\omega$ depends on geography. Runoff increases by a larger fraction in Polarslice compared to Tropicslice world because runoff is most sensitive to $\omega$ when precipitation over potential evapotranspiration is closer to 1 (see for example Zhang et al. (2004) their Fig. 5), as is the case

for Polarslice world. This larger increase in relative runoff causes a larger fractional increase in weathering (decrease in net C emissions) in Polarslice world (Fig. 6D), requiring relatively more cooling (Fig. 6B) to reach a new steady state in the carbon cycle with zero net emissions. The same relative response between Polarslice and Tropicslice world can be seen in the case where $\omega$ is increased, causing a decrease in runoff (Fig. 6E-H). In this case, the decrease in runoff is greater in Polarslice world (Fig. 6G), leading to a larger increase in net C emissions as weathering declines (Fig. 6H).

In both cases (decreasing and increasing $\omega$) Polarslice world takes longer to return to zero net C emissions compared to Tropicslice world, mostly because the runoff perturbation is relatively larger (Fig. 6C,G). We note that the runoff response to warming is muted in Polarslice relative to Tropicslice world, but this effect does not lead to a weaker silicate weathering feedback (Supplemental Fig. S3). Despite the more sluggish runoff response, the temperature response to $pCO_2$ over land in Polarslice world is greater than Tropicslice world due to polar amplification of warming, which increases weathering and

counteracts the effect of a weaker runoff response. The same effect can be seen in Fig. 4, where weathering responds strongly to warming in the northern hemisphere mid-to-high latitudes despite a relatively weak runoff response. Consequently, the weathering response to $pCO_2$ is nearly identical in Polarslice and Tropicslice worlds, despite spatially distinct temperature and runoff responses to $pCO_2$ (Supplemental Fig. S3).

Changes in terrestrial moisture recycling efficiency could have other impacts on weathering that are not captured in our

model. For example, moisture recycling efficiency has no effect on the residence time of soil water in the CH2O-CHOO TRAIN, although this residence time may also modify weathering rates by increasing soil porewater cycling and therefore



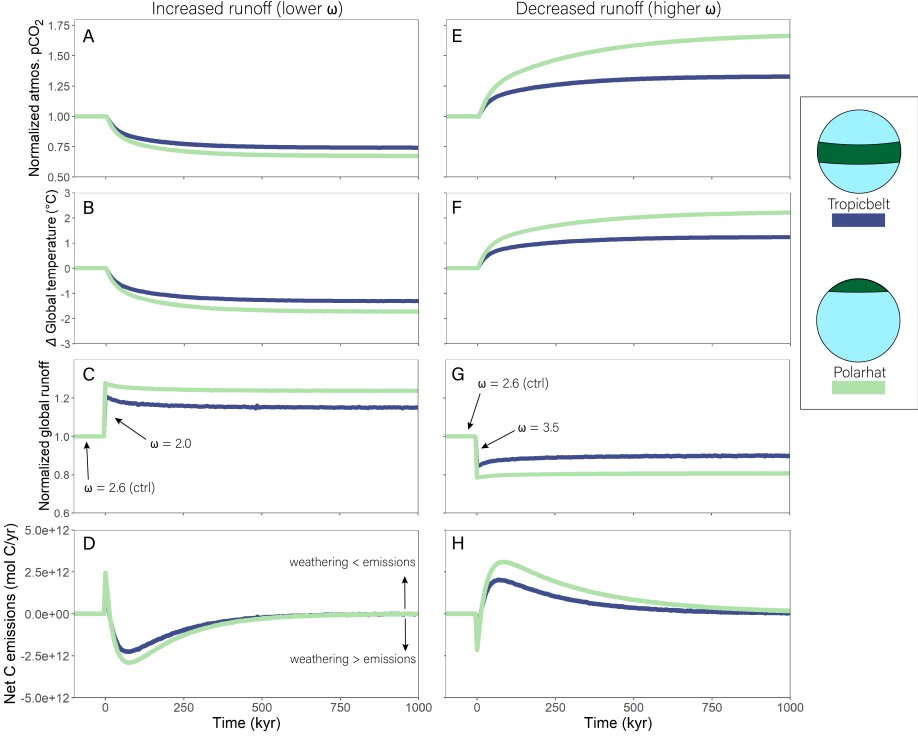

**Figure 6. Abrupt change in evapotranspiration** ($\omega$). Effect of instantaneous decrease in Budyko $\omega$ at time zero from 2.6 to 2.0 (panels $A - D$) on atmospheric $pCO_2$ (**A**), the change in global temperature (**B**), global runoff, normalized (**C**) and net carbon emissions (**D**). Panels **E-H** show the same variables but for an increase in $\omega$ from 2.6 to 3.5. The slower and larger magnitude climate response of Polarslice world compared to Tropicslice world is owed to the larger magnitude change in climate from the change in $\omega$ rather than changes in the strength of the silicate weathering feedback (see text).

riverine solute concentrations (Ibarra et al., 2019). Evaporated moisture is also not given the opportunity to rain out over land further downwind; the CH2O-CHOO TRAIN effectively assumes that any increase in terrestrial evaporation is rained out over the ocean. However, the downwind effect of upwind evaporation often depends on larger scale changes in atmospheric circulation that are difficult to simulate without a more complex climate model (Goessling and Reick, 2011). Thus, our idealized experiments emphasize the role of spatially variable temperature and runoff responses to $pCO_2$ while less certain factors, such as atmospheric dynamics, are ignored.

### 4.5 Instantaneous change in the diffusivity of moist static energy

In our model, moist static energy diffusivity ($D$) determines how efficiently the surplus of atmospheric energy in the tropics and subtropics (where incoming radiation exceeds outgoing) is transported toward the polar regions where there is an atmospheric energy deficit (outgoing radiation exceeds incoming). $D$ has only a loose physical meaning, as it is convenient in models of



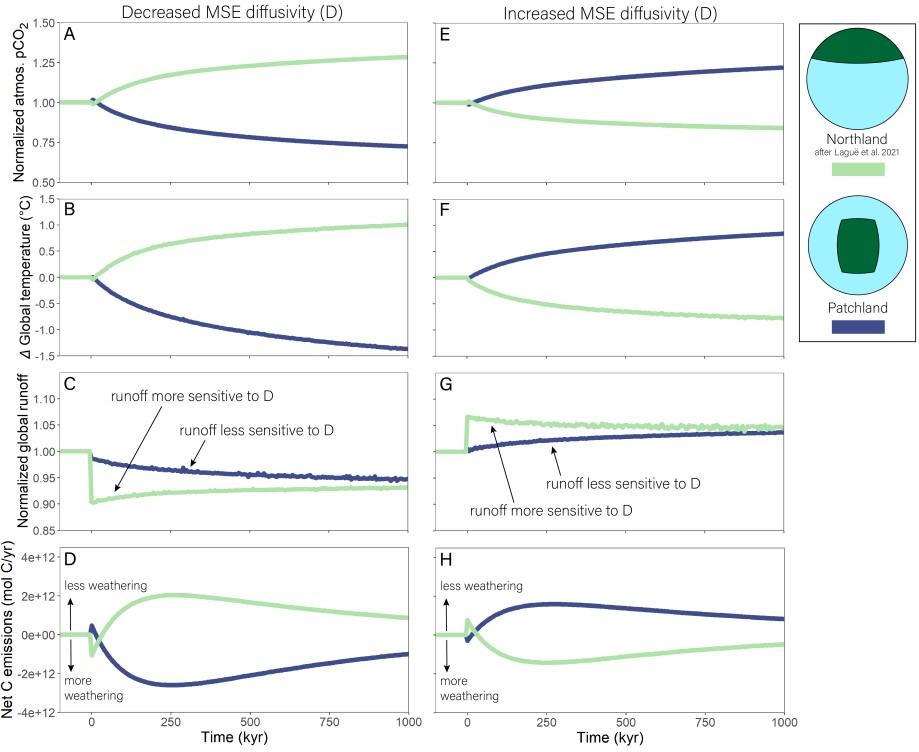

**Figure 7. Abrupt change in MSE diffusivity** ($D$). Effect of an instantaneous decrease in the diffusivity coefficient from $1.06 \times 10^6$ to $0.71 \times 10^6 \ m^2 \ s^{-1}$ at time zero on atmospheric $pCO_2$ **(A)**, the change in global temperature **(B)**, the fractional change in global runoff **(C)** and net carbon emissions **(D)**. Panels **E-H** show the same variables but for an increase in $D$ to $1.41 \times 10^6 \ m^2 \ s^{-1}$. Note that, unlike other variables, the runoff response is not symmetric between the geographies because runoff is sensitive to $D$ in Northland world, but much less sensitive in Patchland world. Simulations are ice-free at all times.

this level of complexity to parameterize poleward moist static energy transport as a diffusive process to subvert the complex, underlying transport physics. Still, the $D$ which produces the best MEBM fit to true climate data is generally thought to change with geography, $pCO_2$, and other factors (Frierson et al., 2006; Peterson and Boos, 2020; Siler et al., 2018). At present, we

lack rigorous process-based formulations for the $D$ response to climate and geography, but nevertheless we simulate its effect on long-term carbon cycling to build intuition for its role in the MEBM and broader CH2O-CHOO TRAIN framework.

Patchland and Northland worlds show distinct responses to the same change in $D$ (Fig. 7). A decrease in $D$ lowers $pCO_2$ and cools Patchland world, whereas the same decrease in $D$ raises $pCO_2$ and warms Northland world (Fig. 7A, B). The divergent climate responses are caused by diverging weathering responses which, in turn, are caused by spatially variable changes in

runoff and continental temperature (Supplemental Fig. S4). Weathering increases with lower $D$ in Patchland mostly due to warming in the subtropics and tropics at the expense of cooling at higher latitudes. Runoff increases in the subtropics but decreases slightly in the tropics such that the effect of runoff on weathering is small. The fact that temperature, not runoff,





drives the increase in weathering in Patchland world is evident in Figure 7C. Here, the initial decrease in runoff in Patchland world is small, and runoff continues to decrease with time as the planet cools. Conversely, runoff and temperature both drive
the weathering response in Northland world. Runoff initially decreases substantially in the polar continent as $D$ decreases (Fig. 7C), then runoff gradually increases as the planet warms and weathering begins to balance emissions. Increasing $D$ leads to essentially the opposite effect (Fig. 7E-H). Patchland world warms due to an initial drop in weathering while Northland world cools due to an initial weathering increase. As in the case of decreasing $D$, the change in weathering in Patchland world is primarily driven by temperature—Patchland world runoff is less sensitive to $D$—whereas temperature and runoff increase in
concert to increase weathering in Northland world. We note that the direct effect of changing $D$ on global temperature is small. Changing $D$ will warm some regions and cool others, with the opposing effects largely canceling out (especially when the poles are ice-free and there is no ice albedo feedback). However, the terrestrial climate response to $D$—which can be sensitive to the continental geography—determines its effect on weathering and therefore long-term global temperature.

### 4.6   Subtropical continents

Runoff tends to decrease with warming in the subtropics in the MEBM module, potentially creating a positive weathering feedback if (1) land is restricted to the subtropics and (2) runoff decreases weathering faster than temperature increases it (see, for example, Fig. 3 and Fig. 4). We test the effect of restricting land to the subtropics on the weathering response by repeating the instantaneous change in moisture recycling efficiency experiments. We test three geographies with "belts" of land with different thicknesses centered on the northern hemisphere subtropics—Business belt, Fashion belt, and Championship belt worlds. In
all geographies, the increase in runoff drives more weathering and negative net C emissions (Fig. 8C, D) which reduces temperatures and weathering rates until balance is restored in the carbon cycle and net C emissions reach zero. Importantly, the decrease in weathering rates (after the initial increase) is driven by different factors in different geographic configurations. In Championship belt world, runoff decreases weakly with cooling, allowing less runoff and colder temperatures to each contribute to decreasing weathering fluxes. In Fashion belt and Business belt worlds, however, runoff continues to increase as
the planet cools. The fact that weathering rates decline and net C emissions return to zero in these geographies indicates that temperature, rather than runoff, is the dominant variable driving weathering fluxes in this experiment.

Conversely, instantaneously increasing $\omega$ (decreasing runoff) causes an initial drop in weathering fluxes, net C emissions, and warming in all three geographies (Fig. 8E-H). The anomalous C emissions are temporary in the broader Fashion belt and Championship belt geographies where some land has a positive runoff scaling and climate reaches a new, warmer steady
state approximately one to three million years post-perturbation. However, in the narrowest belt—Business belt—warming causes a decrease in runoff which further decreases weathering and causes more warming. This runaway hothouse climate in Business belt world reaches $100\times$ the initial $pCO_2$ value by $\sim 2.5$ million years of time (Fig. 8E), with sustained positive net C emissions fating the planet to inhospitable conditions (Fig. 8H).

To understand the distinct climate responses in each geography and each experiment (increasing or decreasing $\omega$) we plot
the normalized global runoff and $F_{w,sil}$ values against the global temperature anomaly in Figure 9A, B. The experiments where global runoff is instantaneously increased are to the left of the vertical gray dashed line, and the experiments where global



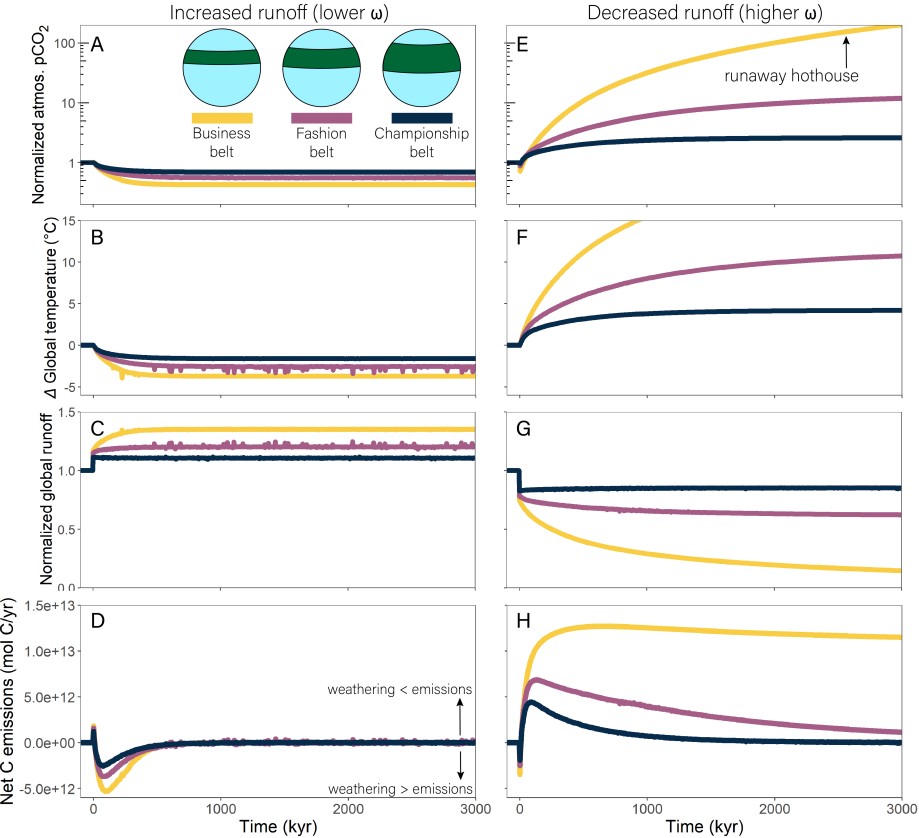

**Figure 8. Changing runoff in subtropical continents**. The effect of increasing runoff on atmospheric $pCO_2$ **(A)**, the global temperature anomaly **(B)**, the relative change in global runoff **(C)** and net carbon emissions where zero indicates steady state **(D)**. Panels **(E-H)** are the same as (A-D) but for a decrease in runoff. The belt of land steadily widens from Business belt world (yellow) to Fashion belt world (pink) to Championship belt world (dark blue). Note that decreasing runoff leads to a runaway climate state in Business belt world while increasing runoff does not.

runoff is decreased lie to the right. In Fig. 9A, we see that the runoff scaling is sensitive to the width of the subtropical land belt. Runoff increases weakly with warming in Championship belt (where the most land extends out of the subtropics), decreases weakly with warming in Fashion belt, and decreases more substantially with warming in Business belt, where land

is most concentrated in the subtropics. The positive runoff scaling in the Championship belt world yields the largest increase in $F_{w,sil}$ with warming of the three geographies (steepest slope in Fig. 9B). Despite the negative runoff response to temperature, $F_{w,sil}$ still increases with warming in Fashion belt world, largely because the effect of warming on weathering fluxes outpaces the effect of runoff. In Business belt world, however, weathering increases with warming in the increased runoff experiment, but decreases with warming in the decreased runoff experiment. The positive relationship between $F_{w,sil}$ and temperature leads

to a negative weathering feedback on carbon emissions, preventing a runaway climate effect (Fig. 9C). In contrast, the negative





$F_{w,sil}$-temperature relationship in the decreased runoff experiment indicates a positive weathering feedback, leading to runaway warming.

The shift from a negative to positive weathering feedback between the lower and higher $\omega$ experiments can be explained by the change in the magnitude of runoff. Silicate weathering fluxes are proportional to the product of solute concentrations
$[C]$ (which increases with temperature) and runoff (which decreases with temperature in Business belt world). The competing effects of $[C]$ and runoff with warming cause this change in the direction of the weathering feedback as very low runoff levels decrease the sensitivity of $F_{w,sil}$ to changes in $[C]$. To confirm that this effect is causing the change in the weathering feedback direction, we build a toy model which uses the temperature sensitivity of $[C]$ and runoff to predict how the magnitude of global mean runoff impacts the strength of the weathering feedback (supplementary text). As global runoff decreases to low values
($\sim< 0.2$ m/yr), weathering decreases with warming as the temperature effect on $F_{w,sil}$ is diminished (Supplemental Fig. S2).

The experiments with our subtropical continents capture some of the unconventional behaviors that can be explored in the CH2O-CHOO TRAIN framework. We note that the weathering feedback in these simulations is only weakly negative (if not positive) compared to most other geographic configurations. As a result, the small runoff perturbations we impose (varying $\omega$ from 2.6 to 3) cause large changes in climate (5-10°C in global warming or more), perhaps suggesting the planet is not resilient
to perturbations when land is concentrated in the subtropics. However, other factors that we do not simulate here—such as a plant-mediated land albedo response to $pCO_2$, or seafloor weathering—could strengthen the negative weathering feedback such that these results are not necessarily generalizable through time. An advantage of the CH2O-CHOO TRAIN framework is that such processes are easy to simulate efficiently, making it possible to test the resilience of global climate to various perturbations when a wide range of processes are turned on or off.

## 5   DISCUSSION

### 5.1   Model application and limitations

The primary feature of our model relative to existing long-term carbon cycle frameworks is the intermediate complexity representation of climate via the MEBM. We expect that the most useful applications of the model will include those analyzing the sensitivity of long-term carbon cycle dynamics to various features of the complex climate system that are difficult to capture
in simpler models and difficult to efficiently test or modify in more complex models. The ice albedo feedback, the role of ice sheets in weathering, polar amplification of warming, and changes in moist static energy diffusivity—processes explored above—are examples of climate features that can impact long-term carbon cycling and can be easily investigated in the CH2O-CHOO TRAIN framework but are often difficult to efficiently manipulate in more complex models. Still, there are important limitations to such model applications that arise from the underlying model assumptions. We emphasize these limitations here
while also highlighting some of the advantages that justify our use of the MEBM and its one-dimensional approach to represent climate.



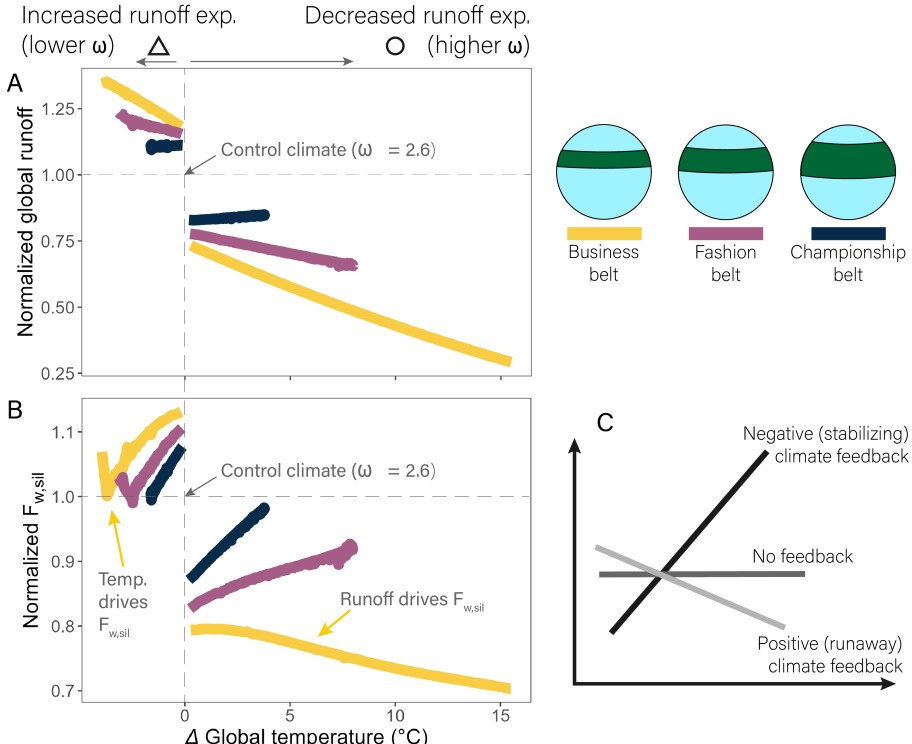

**Figure 9. Runoff and weathering response to temperature in subtropical continents**. **(A)** Runoff decreases with warming most rapidly in Business belt world (yellow), more modestly in Fashion belt world (pink) and runoff increases slightly with warming in Championship belt world (blue) in both experiments. **(B)** Silicate weathering increases with warming in Championship belt world and Fashion belt world (despite the negative runoff scaling in Fashion belt). In Business belt world, weathering fluxes increase with warming when runoff is higher, but decrease with warming in the low runoff (higher-$\omega$) simulations causes a shift to a positive weathering feedback. **(C)** shows a schematic for interpreting the direction of the silicate weathering feedback depending on its response to temperature (axis labels are same as (B)).

### 5.1.1 Climate and weathering

A primary limitation of the CH2O-CHOO TRAIN framework is that the model must be tuned to the desired baseline climate and weathering state. Parameters such as the diffusivity coefficient, land, ocean and ice albedo, climate sensitivity, and others

will affect the climate state for a given $pCO_2$ and will determine the $pCO_2$ thresholds at which ice sheets initiate or collapse. Thus, if the model simulates a shift to an ice-free climate above 700 ppmv, this result should not be considered evidence that 700 ppmv represents a $pCO_2$ threshold for ice melt. Instead, we suggest that users first tune the model to match a desired baseline climate and its sensitivity (as informed by modern or geologic data, or climate model output). In this way, the model is perhaps most useful for studying how key aspects of climate affect the time-evolution of long-term climate and carbon cycling,

and less useful for constraining temperature and $pCO_2$ thresholds of climate transitions which are highly parameterized.





Baseline weathering fluxes in the model are also parameterized using a scaling coefficient to balance silicate weathering and volcanism at the first model timestep. The weathering scaling coefficient, therefore, effectively modifies the strength of the silicate weathering feedback by assigning an implicit slope to the weathering-temperature relationship. The current version of our model also does not include explicitly parameterized seafloor basalt weathering (Coogan and Dosso, 2015). This flux

is sensitive to deep water temperatures, but not to runoff. As a consequence, for most simulations where global temperature and runoff co-vary, inclusion of seafloor weathering is not likely to fundamentally change the results presented here, but will change the timescales over which the Earth system achieves a new stable equilibrium. However, in instances where runoff and temperature are negatively related or unrelated (see Section 4.6), seafloor weathering may act to prevent a runaway greenhouse, though we note that this depends upon the sensitivity of seafloor basalt weathering to temperature.

The scaling coefficient for weathering is important to consider when comparing different geographic settings, volcanic fluxes, or climate states. Changing one of these factors (geography, volcanic flux, or climate) almost always changes another. For example, two different continental geographies with different runoff distributions will require either (1) two different scaling coefficients to match a given volcanic flux; or (2) two different volcanic fluxes for a constant scaling coefficient; or (3) two different climate states for a constant scaling coefficient and volcanic flux. Importantly, this limitation is not unique to our

model framework. In any model for the long-term carbon cycle, it is generally impossible to compare two different geographic configurations, volcanic fluxes, or climate states, while holding all else constant. Changing any of these terms will tend to put the carbon cycle out of balance, requiring compensation somewhere else. Due to this limitation, certain research questions must be approached with caution. For example, the question of whether one continental geography or another yields a stronger silicate weathering feedback is difficult to test because the weathering scaling coefficients, the volcanic fluxes of $CO_2$, and/or

the baseline climate states must differ, all of which may also affect the feedback strength.

However, this formulation of climate and weathering in the model carries distinct advantages, too. Perhaps the most important advantage is that weathering is not explicitly parameterized to increase with $pCO_2$, as is common with low-dimensional box models of the long-term carbon cycle (Bergman, 2004; Caves et al., 2016; Zeebe, 2012). In contrast, higher-order models use climate model data where the temperature and (especially) runoff response to $pCO_2$ is more complex and, in some

cases, weathering has been shown to decrease with warming (Pollard et al., 2013). In our model formulation, the strength and direction of the weathering response to climate mostly depends on the boundary conditions which determine where continental runoff occurs and how it responds to $pCO_2$. Thus, similar to more complex two and three dimensional models, our one-dimensional framework allows for a dynamic silicate weathering feedback which responds to time-variant conditions such as ice cover (Fig. 5) and time-invariant conditions such as geography (Fig. 6 and 7). As a result, it is easy to explore scenarios

that cause or prevent a positive weathering feedback and runaway climate states in the CH2O-CHOO TRAIN framework (*e.g.,* Fig. 8).

Another advantage of our model framework lies in how ice sheets interact with climate and carbon cycling. While the exact role of ice sheets in the long-term carbon cycle remains unclear (e.g. von Blanckenburg et al., 2015; Torres et al., 2017), our model presents a framework to test existing hypotheses in such a way that ice, climate, and the long-term carbon cycle are fully

coupled. This coupling to the long-term carbon cycle via weathering is generally absent in more complex, long-term models



of ice sheet dynamics and climate (Pollard and DeConto, 2005; DeConto et al., 2008), although not absent from all complex models (Donnadieu et al., 2006; Holden et al., 2016; Ridgwell et al., 2007). Zero-dimensional box models have also been parameterized to account for icehouse-greenhouse transitions and their effect on weathering, with previous results showing climate oscillations as ice sheet growth and decay overshoots the equilibrium weathering flux (Zachos and Kump, 2005).

Consistent with more complex models (Pollard et al., 2013), we were unable to replicate this effect in our one-dimensional framework largely because polar weathering fluxes are only a small fraction of global weathering fluxes in most continental geographies.

### 5.1.2    The zonal-mean framework

The key assumption that distinguishes our model from previous one-dimensional energy balance climate models in the long-

term carbon cycle is that the zonal mean climatology produced by the MEBM adequately represents terrestrial (hydro)climate conditions. Indeed, certain assumptions within the MEBM hold only over ocean. For example, we prescribe a spatially uniform relative humidity value of 80%, consistent with oceanic, but not terrestrial, near-surface conditions. We note it is possible to prescribe a spatially variable humidity field, as done in previous work (Peterson and Boos, 2020). Further, the evaporation approximation used in the MEBM (Siler et al., 2018, 2019) is valid over oceans but not land. This is in part due to the fact

that evaporation is not limited by water availability, as is commonly the case over land. As mentioned previously, our approach to translate zonal mean evaporation to terrestrial evapotranspiration involves imposing a water limitation constraint following the Budyko hydrologic balance framework—a step which provides physically reasonable evapotranspiration and runoff values, but which decouples terrestrial evapotranspiration from the zonal mean climatology. While we make efforts to derive realistic terrestrial hydrologic budgets from the zonal mean MEBM results, it is clear that the zonal mean climatology cannot be

considered equivalent to terrestrial climatology everywhere.

Deriving zonal mean weathering rates from the zonal mean climatology can present another challenge. Land surface reactivity can change over space at a given latitude depending on topography, soil age, and other factors (e.g., Maher and Chamberlain, 2014; Waldbauer and Chamberlain, 2005). A landscape with some given mean runoff, temperature, and reactivity will weather more if high runoff and high reactivity co-occur (as in a wet, coastal mountain range with a dry inland plain). Meanwhile, the

same zonal mean runoff, temperature, and reactivity will lead to less weathering if high runoff occurs in a less reactive region (as in a wet, coastal plain with a drier, inland mountain range). While the covariation of temperature, runoff, and reactivity at a given latitude influences the zonal mean weathering rate, this information is lost in our one-dimensional approach.

Still, the zonal mean climatology and weathering remain useful features of our model, even if they are not perfect representations of how the two-dimensional landscape is projected into one-dimensional space. The zonal mean approach is

computationally efficient and makes it possible to consider how spatially complex changes in hydroclimate can impact weathering during a carbon cycle perturbation. For example, if ice melt in the north pole causes the tropical rain belt (ITCZ) to shift north, then the weathering response to this ice melt will depend in part on whether there is more or less land in the ITCZ's new location. The effect of this ITCZ shift would be lost in most 0-dimensional models where weathering and runoff are single functions of temperature or $pCO_2$. Similarly, changes in land albedo are known to shift tropical rainfall (e.g. Charney, 1975;



Claussen, 1997) and can be efficiently represented in this zonal mean framework to explore the carbon cycle consequences. In sum, the zonal mean approach captures critical, realistic processes that lower-dimensional models usually omit while providing more computational efficiency compared to more complex models.

## 6   Next stops

The CH2O-CHOO TRAIN is designed to be computationally efficient and highly customizable, presenting opportunities for
new features, processes, and complexity in future work. For example, terms such as humidity, the diffusivity coefficient, lithology, and rock reactivity are globally constant in the idealized simulations presented here, but could easily be made spatially explicit in the current model framework. We also recognize room for improvement in certain aspects of the model. For example, our current ice sheet formulation is rather crude, with a prescribed ice sheet albedo that occurs whenever temperature drops below a prescribed threshold. More sophisticated ice sheet parameterizations in MEBMs have accounted for other effects
such as ice thickness, sea ice thermodynamics, and seasonal ice formation and retreat (Feldl and Merlis, 2021). The effect of seasonal insolation, specifically, is a feature of interest for simulating tropical weathering where changes in past rainfall often track seasonal insolation trends. Adding insolation seasonality and more complex ice sheet dynamics would undoubtedly increase the computational expense of the model.

The zonal-mean framework of our model is also well-suited for simulating the effect of spatially variable radiative feedbacks
on the climate response to carbon cycle perturbations. While the ice albedo feedback is already accounted for via a temperature-dependency of albedo, other feedbacks such as cloud feedbacks are currently absent from the model. However, the zonal pattern of such feedbacks could easily be prescribed, perhaps from climate model output, to explore how the effect of these feedbacks on temperature and hydroclimate impact the time-evolution of weathering (Roe et al., 2015). Still, the suite of feedbacks in the current model, including the combination of radiative and weathering feedbacks, are rarely considered in a single model
framework. The CH2O-CHOO TRAIN therefore brings opportunities to explore the complexity that emerges through the myriad interactions between climate and the long-term carbon cycle in the geologic past.

*Code availability.* The code, instructions for running the model, and associated scripts for plotting model output and generating model input files can be found on Github (repository: https://github.com/tykukla/CH2O-CHOO-TRAIN) and Zenodo (Kukla et al., 2022). The code used for the analysis in this manuscript is tagged as release v1.0.

*Author contributions.* All authors conceptualized the model and contributed ideas and code associated with model construction. TK led the model integration and testing. All authors contributed equally to writing the model formulation section of the manuscript. TK wrote the rest of the manuscript and all authors contributed to reviewing and editing. TK designed and conducted the analyses in the text and formatted the model code for distribution.





*Competing interests.* The authors declare no competing interests.

*Acknowledgements.* We gratefully acknowledge Nick Siler and Gerard Roe for guidance on coding the energy balance model and under-
standing its strengths and limitations as they apply to paleoclimate problems. JKCR acknowledges postdoctoral funding from the Alexander
von Humboldt Foundation. DEI acknowledges postdoctoral funding from the UC Berkeley Miller Institute and UC President's Postdoctoral
Fellowship. KVL acknolwedges partial support from the Agouron Geobiology Postdoctoral Fellowship. This is EHGoS contribution no.
17521.



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
