# Peer review of "All aboard! Earth system investigations with the CH2O-CHOO TRAIN v1.0"

_EGUsphere, 2022_

## Referee Comment (RC2)

**Review of "All aboard! Earth system investigations with the CH2O-CHOO TRAIN v1.0"**
by
Tyler Kukla, Daniel E. Ibarra, Kimberly V. Lau, and Jeremy K.C. Rugenstein

**Summary and general comments:**

Kukla and colleagues present the new computationally very efficient Earth system model framework, CH2O-CHOO TRAIN, that combines three existing modules: i) a zonal mean Moist Energy Balance Model (MEBM; after Roe et al., 2015), ii) a continental weathering module (after Maher and Chamberlain, 2014), and iii) a simple box model for the long-term carbon cycle (after Caves Rugenstein et al., 2019). The MEBM can be configured with different geographies to investigate its effect on temperature and runoff distributions for global weathering and carbon cycle dynamics. Especially due to the simple carbon cycle representation, the model simulates about 1 million model years in thirty minutes on a standard laptop. As such, it fills a gap between highly parameterized conceptual box models and Earth system Models of Intermediate Complexity (EMICs); thus, it represents a valuable new model for understanding interrelated Earth system dynamics.

The low computational demand of CH2O-CHOO TRAIN makes it very useful for large-ensemble experiments that are needed for uncertainty quantification. Another advantage the authors mention is that their model is "highly customizable", i.e., "making it easy to directly modify processes in the climate system". Unfortunately, both model strengths are not very well exploited in the manuscript. The model behavior is not compared for the different continental configurations, and only a few parameters are changed to lower and higher values. Why did the authors not put more work into investigating how the TRAIN really behaves, e.g., under different perturbation scenarios and/or model configurations (i.e., investigating structural uncertainties)? Many processes are not explicitly represented in the model and are thus highly parameterized and a large source of uncertainty. Therefore, a comprehensive sensitivity analysis is needed to quantify this uncertainty, identify the most sensitive parameters and adequately understand how the model works (see e.g., Pianosi et al., 2016, 2015, for some methods and ideas). Because this is the first CH2O-CHOO TRAIN model development paper and the model runs so fast, a comprehensive sensitivity study is needed and feasible.

Some parts of the manuscript are not well explained (motivation of the experiments) or information is missing (e.g., a comprehensive table stating model parameter names, values units, references; organic carbon burial & weathering; isotopic balance of the system) or confusing (e.g., is S and P calculated by the weathering module: if yes, how and why?).

Overall, I think the model can be a valuable tool to understanding Earth system dynamics and past climate variations which crucially depends on the combined use of different numerical representations of the Earth system. However, the manuscript would benefit from some more work to help the model uncertainty better and to showcase applications of the CH2O-CHOO TRAIN.

**Specific comments:**

Abstract: Please shorten some parts of the background and include a few main model results presented in the ms.

It could be made more clear what the main improvements of the model are compared to previous approaches, like COPSE or GEOCARB: I suppose it is the 1D atmospheric energy balance model and the improved water cycle? And also the possibility to represent a continental configuration for the MEBM model?

Related to this: be more specific how the climate processes in the model can be modified. This is mentioned e.g. in the abstract, introduction (~line 45) and the start of Section 3 (line 369). The authors are the most specific when they mention "The model is designed to be highly customizable, making it easy to directly modify processes in the climate system such as ..." (line 44-45) and that the processes are either highly parameterized in simpler models or emerging properties in more complex models. I understand that this is a major strength of the model therefore it would be good to discuss this in more detail, give examples what specifically can be changed and how (I suppose the parameters changed in your experiments are not all possible parameters). A table summarizing what parameters can be changed to affect the different processes in what direction might be helpful.

In the very beginning I thought the continental configuration would also impact the Long-term C-cycle but it actually does not. That should be made more clear in the text and in Fig 2.

**Figure 1:**
It was not obvious to me how some components shown are necessary to initialize the model. E.g., it is unclear how knowledge about lithology, soil age, soil pCO2 influences the weathering model (I am not able to judge the MEBM part – this is not my expertise). I did not see parameters for them in the main manuscript. E.g. soil age ($T_S$) is given in Table S2 but I can't see an equation where it is used – or it is $t_{wz}$ in equation (12).

**Model**
The main manuscript does not include a table stating the main model parameters, their values, units and references. This makes assessing the model construction difficult. Some are given in the SI – this could go into the main manuscript. But do the Tables S1 – 3 include all important model parameters? Please make sure the units for all model outputs are stated, e.g. no units are given for $q_{land}$ and $F_{w,sil,carb}$ after equations 8 + 18 or what is meant by [C] (+ units) in line 195 is not 100% clear.

Also, I suppose various model parameters are very uncertain. Therefore, a sensitivity study which assess the importance of these parameters for different model outputs and calculates quantitative sensitive indices would be very informative in order to understand how the model behaves and its results. In line367 – 368 you say: "These experiments are not meant to be an exhaustive sensitivity analysis of the model." But this is the first version of the model, therefore, in my opinion, a more complete sensitivity analysis is necessary. In contrast, the often long and sometimes vague discussion of model limitations (Section 2.5 & 5.1) could be shortened.

Simulating modern conditions:

It would be very informative to provide a more in depth evaluation of the model results with modern boundary conditions – which, I think, should be the main reference simulation to establish that the model works well. E.g., you could compare your model output with other estimates and/or observations (e.g., plot this in Fig. 4 A – C and also the simulated E minus P). Do I understand it correctly that the output of the C-cycle model is prescribed for the steady-state condition, i.e. all the initial fluxes in Table S3?

**2.1 Moist Energy Balance Model** – I don't have the expertise to review this in detail.
But where does the temperature threshold of -5°C for the appearance of ice come from? And should this not be very different for ice sheets (on land) and the formation of sea ice? Please give a reference and justify.

How is temperature calculated in the MEBM?

What happens to terrestrial runoff over the ice sheets? Should this not be significantly reduced because the precipitation is snow and becomes part of the ice sheet? This might not be important for the default setup of the model because it does not affect weathering but for your experiments "effect of ice cover on weathering" I suppose it is.

**2.2 Weathering:**
lines 239 ff. "We parameterize maximum carbonate weathering reaction rates as being 1000 times faster than silicate weathering… " Is this a reasonable assumption? I would assume these parameters have large uncertainties and a large effect on the model output. I would suggest include these (and similar parameters) in a sensitivity study.

Please provide information for how organic carbon weathering is calculated? Is it constant?

In the text (line 101) and in Fig. 1 the authors mention that the weathering module calculates fluxes of P & S, but this is not discussed/introduced in the model description. Why is this even in the model? If I understand it correctly, your ocean PP is not simulated (Corg burial is only scaled to CaCO3 burial) – hence P is not needed. And I don't understand why a S-cycle would be needed ? There is no information in the manuscript!

This section should also include details for the isotopic signals of the weathered Corg and carbonates – some of it is given in Table S3 but it would be good to include it in the main manuscript.

Table S3 does not state the initial silicate weathering rate. I suppose it is equal $F_{volc}$?

**2.3 Carbon cycle:**
Please state how organic carbon burial is scaled to CaCO3 burial? Is it done as by Kump & Archer (1999) to isotopically balance the system?

Steady-state of the model:
Related to the last comment: To achieve steady-state is it also necessary that organic carbon burial equals organic carbon weathering – or is this always the case in the model? Or do not have the option to restore some of the buried Corg via volcanic outgassing (see e.g., Lenton et al., 2018, Table 2)?

Please give information how the system is isotopically balanced! This is not discussed.

**2.4 Coupled climate-carbon cycle model initialization and integration**
Line 299: what does 'the carbonate speciation described above' mean? I suppose, you recalculate it every timestep, right – and also $W_{carb,sil}$? Is pH also updated? How?

Line 300: What is meant by "temperature guesses"? Why do you need to guess? Also the rest of the paragraph is unclear to me – why does the model result in a snowball? Does that mean the temperature is everywhere below the threshold of -5°C? And what does "In this case," (line 301) refer to?

**2.5 Model assumptions and limitations**
This part, for the most part, speaks about model limitations. And could therefore be removed from Section "2 Model Formulation" and go to the end of the manuscript and combined with "5.1 Model applications and limitations"

**3 Model Experiments**
The different experiments are not very well motivated or explained. This could be expanded and subsections could be used as in Section 4. A table summarizing the setup of the different experiments would be very helpful.

Please state clearly what the background climate & C-cycle state is for set of experiments? For Fig. 6 – 8 only the normalized values are given – without information how they are normalized.

Would it not be informative to do the same perturbation experiment with every continental configuration to evaluate the effect of geography?

Especially, the description of the second series of experiments using the "Northland" geography is unclear. What parameter are the authors changing? And what is the effect: How much weathering happens under an ice-sheet? Or also the amount of runoff? This is not clear from Section 3. Maybe that is why I struggle to understand the model results (see comments on Section 4.3).

Subtropical continents:
I am not entirely sure how the last set of experiments links to the hypotheses of the breakdown of the silicate weathering feedback. You arbitrarily change the runoff in an experiment that is at steady-state which then causes the feedbacks to respond. And yes, you get a runaway greenhouse for a small continent where the runoff is very sensitive to changes in $w$ and the stabilizing strength of the silicate weathering is weak. Would it not be better to to setup different steady-state experiments with the three configurations (and different values of $w$) and then perturb the system (i.e., by injection of CO2) and see how the responses are different?

**4 Results**

**4.1 Reference Simulation**

Here you talk again about the temperature guess. This is unclear to me.

Line 424-426: "If both temperature guesses are the same, the first pole to glaciate in the meridionally symmetric case will depend on the tuning of the numerical solver."
Can the authors please give more details here. Maybe this is related to my question how temperature is calculated in the MEBM?

**4.2 Response to abrupt pCO2 increase with modern geography**
What is the isotopic signal of the carbon injection?

The motivation given for this experiment is (line 376): "This simulation is used as a verification of the coupled model's performance in comparison with other, similar simulations across the model hierarchy." This is not done at all.

There could be more content in this section. The model runs so fast. Why do the authors not put more work into investigating how it behaves under different perturbation scenarios or model configurations or for different parameter settings. I feel like this is a missed opportunity to understand their model better.

Unclear how the last two sentences of the section fit in.

**4.3 Varying the effect of ice cover on weathering**
If I understand the experiment correctly two parameters are changed at the same time:
1. % of effective runoff
2. volcanic outgassing
I think this makes it difficult to disentangle what's going on. Imagine, keeping volcanic outgassing unchanged and only changing the effective runoff: this should already result in a different equilibrium climate because some weathering is possible under the ice sheets. If I am correct, this should be evaluated first by the authors.

I find the text difficult to follow. In general, it might be good to start from your default setup: which is, as I understand it, is 0% effective runoff. And then describe what happens if effective runoff is increased.

Please plot the global area of ice-cover as it is talked about in the text, it's a main part of the experiment and important to understand what's going on. This might also solve my confusion with the statement in Fig. 5 "Ice sheet growth limits runoff": Why would this be largest in the experiment with the warmest climate? Ice sheets should expand the least here.

Also what about the ice-albedo-temperature feedback? Does it not play a role for the results of Fig. 5? It is only mentioned at the end to explain the step-changes.

Why does the model calculate higher mean runoff in the 100% effective runoff setup which is the colder climate state. I had the impression runoff scales mainly with temperature (see, e.g. Fig. 4E, F).

**4.4 Instantaneous change in moisture recycling efficiency**
Why are all results now shown normalized and how is this done?

Fig. 6 D: Why is there first an increase in net C emissions? This is never discussed.

In my opinion, the last paragraph (lines 494 - 502) does not belong into the results section. If at all it could into the limitations section.

**4.6 Subtropical continents**
line 530: "Runoff tends to decrease with warming in the subtropics in the MEBM module" can you show model output for that? It is in contrast to the statement in lines 436 – 437: "runoff is generally insensitive to global climate between 30 and 50 degrees latitude".

The description of Fig. 9 is very confusing as it jumps back and forth between increased runoff and decreased runoff experiments and explains how runoff changes through the transient experiments.

Isn't the small continent of business belt world a big factor why it runs into a runaway greenhouse: i.e. the weathering feedback is weaker than in the other worlds and runoff is probably also more sensitive to changes in *w*.

**5 Discussion**
It is not really a Discussion paragraph. Maybe something like "Scope of applicability and limitations" would fit better.

The numbering is a bit odd. Why do you need 5.1 if there is not a 5.2?

Maybe move "2.5 Model assumptions and limitations" to the end of the manuscript and combine with information given in "5.1 Model applications and limitations". Or it might be useful to have two different sections: 1) Model applications  2) Model limitations

**Conclusions**
The manuscript is missing a conclusion paragraph.

**Technical corrections:**

Line: 34: 'the most physically realistic'  maybe better write 'a more mechanistic'

line 43: add:  and 'a' box model for …

line 65:  Maybe '… account for more spatial dynamics than the 0-D representations ...'  because it is just 1D so does not account for all the spatial dynamics

line 68: "precipitation (assumed proportional to runoff)" The phrasing is a bit unclear to me.
Do you mean runoff is proportional to precipitation (because it sounds like the model calculates precip first)? And why is this needed here? Because you are eventually interested in runoff?

Line 73: maybe say 'in a more mechanistic way'

line 97: delete the first occurrence of the word 'box'

line 100: add 'long-term carbon cycle **model**' +  singular for 'outputs'

line 101:should read: These fluxes are used **to** calculate

Equation (7) *w* is not defined here. In Figure 6 you call it evapotranspiration. Which is confusing because ET was defined as evapotransipiration after Eq. (7).

Equations: be consistent with commata in subscripts, e.g., compare (19) vs (21)

line 195: It might be good to explicitly state here: "We calculate solute concentrations for inorganic carbon, [C], ... " To make clear that all forms are considered here.

line 281: please provide a reference for the 10° colder also in the text – not just in Table S3

line 363: small letter p in phosphorus

Missing references for PETM C injection: lines 375-376, 449-450

line 420: why geography? You are only looking at the cat-eye geography here!

Line 471-473: this is trivial

Fig. 6: And related text, define what net C emissions are.
The names of the geographies are different in Fig. 6.

line 636 + 637: Please be careful: GENIE (i.e., Holden et al., 2016; Ridgwell et al., 2007) does not include a representation for ice sheets but only a sea ice model.

Fig. S4: has the wrong exponents for the *D* values

**References**
Caves Rugenstein, J. K., Ibarra, D. E., and von Blanckenburg, F.: Neogene Cooling Driven by Land Surface Reactivity Rather than Increased Weathering Fluxes, Nature, 571, 99–102, https://doi.org/10.1038/s41586-019-1332-y, 2019.

Maher, K. and Chamberlain, C. P.: Hydrologic Regulation of Chemical Weathering and the Geologic Carbon Cycle., Science (New York, N.Y.), 343, 1502–4, https://doi.org/10.1126/science.1250770, 2014.

Pianosi, F., Sarrazin, F., and Wagener, T.: A Matlab toolbox for Global Sensitivity Analysis, Environ. Modell. Softw., 70, 80–85, https://doi.org/10.1016/j.envsoft.2015.04.009, 2015.

Pianosi, F., Beven, K., Freer, J., Hall, J. W., Rougier, J., Stephenson, D. B., and Wagener, T.: Sensitivity analysis of environmental models: A systematic review with practical workflow, Environ. Modell. Softw., 79, 214–232, https://doi.org/10.1016/j.envsoft.2016.02.008, 2016.

Roe, G. H., Feldl, N., Armour, K. C., Hwang, Y.-t., and Frierson, D. M. W.: The Remote Impacts of Climate Feedbacks on Regional Climate Predictability, Nature Geoscience, 8, https://doi.org/10.1038/NGEO2346, 2015.

---

## Author Response (AR1)

Reviews and responses

Kukla et al. present here a model of global geochemical cycles (though the manuscript focus on carbon cycle) and climate at multi-million years timescale. They justify their approach among others by the integration of a recent improvements in the representation of hydrological cycle in energy balance models (Siler, 2018, 2019), which is a novelty in Earth system models of that level of complexity. They provide an adequate discussion of the advantages and limitations of the model framework (though some further limitations can be touched upon). One important advantage being the compromise between representing some key features of the interactions between climate dynamics and continental fluxes, while being able to simulate long timescales needed to investigate geochemical cycles. This is particularly relevant if one consider further developments (or "stops") concerning elements with long residence time, such as oxygen and sulfur. The authors also provide an appreciable discussion of the model's potential applications, and applications for which it is not optimized. The presented experiments give a good illustration these applications, and of the model's behavior.

The manuscript suffers from several incorrectnesses and missing information that the authors need to address. Apart from these, I found the model well described and its results well explained.

Therefore, I recommend minor revision.

 Thank you for the thoughtful and thorough review!

**Main comments**

* There is one key missing element in the description: "$r_{max}$" (Eq. 12) is scaled to "$k_{eff}$" (line 280 of both Initialization/Code/CH2O-CHO_p_func_bistable-trackBC.R and Initialization/Code/CH2O-CHO_p_func_bistable-trackBC-INSOL.R), and not held constant, as stated by the authors (lines 207–208). This scaling was, indeed, already put forward by Maher & Chamberlain (2014).

It is critical because, without this scaling, the only effect of temperature is to reduce the weathering zone reactivity ($f_w$ in Maher & Chamberlain, Eq. 12). So weathering rate would be decreasing with temperature, whatever the value of $pCO_2$ and runoff (if held

constant). This would contradict all the discussion about weathering sensitivity to temperature and runoff.

The authors need to update Eqs. 12 and 13 (and Table S2), also to avoid confusion between $k_{eff}$ and $r_{eff}$.

Thank you for catching this! This comment helped us find an error in the code with how we handled keff (basically, the temperature dependence wasn't getting passed properly to reff). We have modified the code, repeated all simulations (including other fixes discussed below), and clarified the terms in the text. We also added an equation (now equation 15) to make explicit how reff (now rmax to avoid confusion) is scaled to keff.

* The ability and limitations of the model to simulate the hydrological cycle in the various experiments can be more discussed. Siler (2018) showed this current MEBM reproduces the link between hemispheric asymmetries in $Q_{net}$ and the position of the ITCZ and the associated cross-equatorial heat transport. It will be meaningful to show how well (or not) the Northland experiments reproduce the southward shift of the ITCZ observed by of Laguë et al. (2021) in NorthlandBright configuration.

We added this result to the supplement (see also main text lines 418-420) – Northland simulations in the MEBM reproduce the south-shift of the ITCZ of NorthlandBright with approximately the same magnitude (about 5 degrees latitude shift). The annual mean shift in Laguë et al. is difficult to discern because monthly data are reported, but NorthlandBright appears to cause a ~6 degree southward ITCZ shift per their Fig. 10e.

The authors indicate that the E-P formulation "captures the general trends on land". While this statement is supported with modern continental configuration, they omit to mention that the Budyko framework assumes a limitation of evaporation by available water on land (precipitation), but no limitation of precipitation by the distance to the ocean (continentality). In that regard, the Northland configuration, with almost an entire hemisphere continental, pushes the model towards the edge of its domain of validity. The hydrological cycle of Northland here probably resembles more the one of ThreePatchLand configuration from Laguë et al. (2021), where water bodies help carrying moisture up to the North pole.

Thanks for this point, we've added text about this caveat to the model limitations section (see lines 651-659).

It could also be reminded in the discussion that the hydrological cycle is, to some extent, constrained by the imposed profiles of w, u and R_G, meaning that some

features of climate changes under different continental configurations, or pCO₂, may not be seen. Is there a reference for the idealized profiles of R_G and u? It seems to be a determinant choice regarding the hydrological cycle.

We've clarified that the profiles are based on Siler et al. 2018/19.

Finally, a substantial element of this article is the suggestion that silicate weathering turns into a positive feedback if land is concentrated between 10° and 30° of latitude, because of the "arid" behaviour of drier conditions when global temperature increase. On modern continental configuration, most of India and South-East Asia, including the Himalaya, is between 10°N and 30°N. Yet, they don't fall into an arid climate zone, and the reason for this is monsoon. One should expect that the concentration of land between 10° and 30° of latitude will generate a monsoonal circulation, that cannot be captured with this MEBM. The authors need to touch upon this potential challenge to their weathering feedback breakdown hypothesis.

Thanks for this point – we have deleted this section. Our original goal was to demonstrate that a negative weathering feedback is not forced in the model (it's emergent), but we agree this section was more confusing than helpful in making that point.

* The model description is missing one equation to describe $Q_{net}$ and connect it to $LW_{out}$ (Eq. 9). Siler (2018) computed Qnet from modern climate reanalysis (or climate simulations), and used it directly as a forcing. However, it is not what the author are doing (in Initialization/Code/MEBM_ODEfun.R), they compute $Q_{net}$ as "(1-albedo)$I$ – $LW_{out}$", "$I$" being the insolation, which is the forcing term.

We added this equation as well as the default insolation equation.

Following that, I recommend to change the description "difference between top of atmosphere (TOA) and surface net downward energy fluxes" (line 115). Given the construction of the model, there is no net surface flux, so $Q_{net}$ is simply the TOA net downward energy flux.

We have changed the phrasing.

Finally, the author only mention insolation line 421, for the "reference" ("Cat-eye") simulation. I suppose that the insolation forcing the same for all the other experiments?

Yes, this is now clarified in the text where we added the default insolation formulation.

* The authors never explain what is "net C emission". I suppose that it is the sum of all fluxes in Eq. 19, but it should be specified.

Thanks, this is now clarified (line 523-525)

Following that remark, it is not easy to follow what the authors mean by "initial increase/decrease/drop of weathering" throughout the whole section 4. Lines 522–523 gives a good example: "Northland world cools due to an initial weathering increase". Yet, looking at Fig. 7H, what one can see is, at the time of the perturbation (increased $D$), a peak of positive net carbon emission (so towards the "less weathering" direction) for Northland. This contradiction is found in nearly all the experiments.

Assuming that "net C emission" is indeed the sum of all fluxes in Eq. 19, the "very initial" positive peak in net C emission is likely due to increased carbonate weathering bringing more C to the system (whereas silicate weathering has no instantaneous C effect strictly speaking). Only after the ocean chemistry respond to the perturbation can the increased alkalinity flux translate into negative C emission through carbonate burial. The author should indicate that the "very initial" instantaneous response is due to uncompensated changes in carbonate weathering, and is always neglected (justifiably so) in the interpretation. What the reader need to focus on is the "second initial" increase or decrease following the instantaneous response.

* How is weathering of organic carbon computed? Lines 354–355 suggest that it is constant, but it could be more explicitly said. If it is indeed constant, while organic carbon burial is scaled to carbonate burial, I suspect that there is a net imbalance of organic C cycle at the end of all simulations: since carbonate weathering does not have the same sensitivity to climate than silicate weathering, $F_{w,sil}$ getting back to its initial value ($F_{volc}$) – as climate evolves towards its new equilibrium – does not necessarily mean that $F_{w,carb}$ goes back to its initial value. Consequently, neither would $F_{b,carb}$ and $F_{b,org}$.

Is that imbalance small enough to be neglected?

Thanks, this is an important point – there can be imbalances in organic carbon cycling. We now note this in lines 384-386. The imbalances are generally too small (<5%) to have a notable impact on atmospheric pO2.

* The model resolution (discretization) is not given.

Resolution was previously noted in lines 95-97 (now lines 100-101) which stated "(1-dimensional; ~ 200 km resolution) ... (run at 5 kyr timesteps)". To further clarify the

discretization, we added that there are 100 equally-spaced grid cells from pole to pole (though discretization can be user defined).

* The units of land area in Fig. 4, discharge in Figs. 4B, S4C and G, and silicate weathering in Figs. 3C, 4C and S4D and H, are confusing. It is unclear whether the authors plotted the extensive properties (m2, m3/yr, mol/yr) for each latitudinal element (or grid cell) of the model, or whether they plotted their density functions per units of $x$ (which happened to be unitless). In the first case, the values shown in the y axis would depend on the model's discretization, and if the discretization was not regular in $x$, latitudinal variations on those figures could be simply due to grid cells having different area.

I recommend to specify on the figure "density of water land area/water discharge/silicate weathering per unit of $x$" (provided that it is indeed what they plotted), or to simply show the land fraction (continental fluxes are trickier because specific fluxes, like mol/m2/yr, hide the modulation by the land fraction).

Indicate also on those figures the latitudinal scale, which seems to be equal-area ($x$).

With those 2 conditions, the area under the curve will truly represent the Earth-integrated flux, regardless of the chosen discretization.

We clarify in the captions that the x-axis is equal-area, and the values plotted reflect each latitudinal grid cell. We also now plot silicate weathering per km2 in Figure 3 to show the spatial pattern of the rate (independent of land area) whereas we plot the flux in Figure 4 (which captures the effect of land area).

Finally, the units "% of global discharge" (or "% of global weathering" on Fig. S4D and G) suggests that the flux is normalized by the current Earth-integrated flux, and the curves should always sum to 100%. Yet, discharge on Fig 4B is virtually everywhere higher at 2360ppm than at 320ppm. Similarly, weathering seems everywhere lower (higher) on Fig S4D (H) if we compare the light green to the dark blue curve.

We clarify that we plot the percentage of global discharge relative to a baseline case (the purple line in each set of plots).

**Mathematical mistakes**

* Eq. 7 is wrong, the rightmost part should be "$-E_0/P + [1 + (E_0/P)^\omega]^{(1/\omega)}$".

This mistake is not in the code (line 86 of Initialization/Code/MEBM_solve-bistable-glwx.R): the final "-1" of Eq. 7 multiplies the preceding part, instead of being added to.

Thanks for catching this – we have made the correction.

* There are some inconsistencies with Eq. 6 (Supplement Eq. 4).

It should be "$c_P/(L_v\,q)$" (The code is OK: "cp/Lv/q", line 69 of Initialization/Code/MEBM_hydrofun.R)

Thanks, corrected.

As it is, evaporation is expressed in W/m2, which could be indicated, given that $E$, $P$ and runoff are after discussed in m/yr.

We added a note about this after equation 8.

Finally, Siler et al. (2019) derived this equation using $q*$ (saturation specific humidity), which would be $q/rh$ here. Is there a reason why the authors used $q$ instead?

Thanks for catching this too. This was an artifact from model testing and has been corrected (along with updating our simulations). The effect is very small since RH is large and globally uniform.

* I don't understand the justification for the "last term" ($pCO_2 - pCO_{2,0}$) of Eq. 15.

Eq. 15 without the last term (which is the original equation from Volk, 1987), can be rewritten in:

$WZ_{CO2} = pCO_2 + R_{GPP}*(WZ\_{CO2,0} - pCO_{2,0})$

This already ensures that $WZ_{CO2} > pCO_2$ , unless the reference one ($WZ\_{CO2,0}$) is not.

I don't expect, however, this modification to bring any significant change, since the $pCO_2$_soil-$pCO_2$ relationship looks pretty similar with or without the extra term.

Thanks, this is correct and the change has been made (and incorporated in updated analyses, the effects are very small).

* Eq. 20: There's a sign mistake here, it should be "$(\delta^{13}C_{flx} - \delta^{13}C)$" in all terms ("flx" being "volc", "w,org", ...). Same thing line 285, at least given how it is written in Eq. 20.

Thanks, corrected.

This mistake is not in the code (lines 596 and 802 of Initialization/Code/CH2O-CHO_p_func_bistable-trackBC.R and lines 604 and 804 of Initialization/Code/CH2O-CHO_p_func_bistable-trackBC-INSOL.R)

**Details**

I recommend adding a table synthesizing the World configurations, instead of the information being dispersed in the whole section 3.

Thanks, this helped streamline much of this background. The table is added to Figure 2.

Moreover, there's no information about the position of "tropicslice" and "polarslice" (though the figures and text suggest that they are, respectively, centered on the equator and reaching the North pole).

This information now appears in the table in Fig. 2

There is no mention either of the latitude boundaries of "Patchland", the value of the proportion of land of "Patchland" and "Cat-eye" and whether this proportion of land is constant in "Patchland" (as it appears to be on Fig. 2).

Also now in the table in Fig. 2

Finally, the author never present the configuration "Polar[slice|hat] XL", from Supplementary Fig. S4.

A relict from an older, more confusing naming convention – this is updated to "Northland".

"Polarslice" and "Tropicslice" have become "Polarhat" and "Tropicbelt" on Fig. 6, Fig. S3 and its caption

Thanks for the catch – another former naming convention issue that is now fixed.

- Line 101: typo: "used TO calculate"

Thanks, fixed.

- Line 114: "$F$" is the northward flux, not the divergent flux. "d$F$/d$x$" is the divergence.

Also, it would be more accurate to describe $F$ as the column-integrated AND zonally-integrated northward energy flux (which is consistent with its units being W).

Correct, thanks – this has been fixed.

- Line 120: units of $c_p$ is "J/kg/K", not "J/kg"

Thanks, fixed.

- Line 121: It should be less confusing to introduce "$q$" in kg/kg, since it is multiplied by "$L_v$" in J/kg.

Agreed, fixed.

- Line 158: Don't you mean "$E$ is limited by $P$"?

Indeed, fixed.

- Line 178 (Eq. 9) and Supplementary Table S1: the meaning of "$B$" seems to be the Planck's negative feedback, that is, Earth emitting more longwave radiation as its temperature increase. It takes into account the water vapor positive feedback, as the value of $B$ is lower than the pure blackbody emission, but it cannot be described as "Water vapor feedback coefficient".

Thanks, fixed (now described as Planck feedback sensitivity coefficient).

- Lines 189–193: The authors could refer to the Supplement (section 2.3) here.

Added.

- Lines 212–213: This is confusing. "$k_{reac}$" doesn't seem to encapsulate the effects of mineral surface area and molar mass, since they are explicitly separated from this parameter. Besides, its units is mol/m2/y, not y^-1.

Fixed with the corrected formulation.

- Line 407: There's a misstatement here, Kump (2018) does not argue about a positive weathering feedback, they suggest an inefficient (but still negative) weathering feedback because of weathering being limited by the amount of exhumed material.

Yes, thanks, we removed this reference.

- Lines 441–448: Elevation is another key contributor too colder temperature in the South pole, because of lapse-rate and because it favors ice-sheet inception.

Good point – added.

- Section 4.5: A word of caution should be added: changing $D$ imply changing the Hadley cell and eddies-driven transport of moist static energy without changing their relative contribution ($w$). The choice of this sensitivity test therefore have consequences on the tropical/subtropical runoff response.

Thanks for pointing this out – we added this note to the model experiments section (line 436-437)

- Line 565: There is a simplification here: solute concentration [C] increases with temperature, but also decreases with runoff.

This section was deleted.

- Lines 710–720: Another element that could be mentioned here is that designing experiments with constant F_volc and adjusted W coefficient – rather than the opposite – is equivalent to keeping constant the residence time of C, and therefore, the equilibration time of C cycle.

Thanks, added.

- Fig. 1: Does "S" refer to sulfur, that is also simulated in the full geochemical model?

We removed the S.

- Fig. 2: the "Northland" cartoon is confusing because, compared to "Tropicslice" and "Business belt" just next to it, it seems that lands start around 30°N, instead of 12°N stated in the text. I suggest to redraw that cartoon.

This is accurate (Northland begins at 40N), the text has been changed to reflect this.

- Fig. 5: Though it is mentioned in the text, I suggest to specify the geography (continental configuration) on the figure, or its caption.

Added.

- Eq. 13: "$k_{eff}$" from previous equation and text has turned into "$r_{eff}$". Also, these equations (12–13) need to be update to describe the scaling of r_max (see my first comment).

Done.

- Eq. 16: While being exactly how Volk (1987) introduced the equation and its parameters, I found it oddly formulated, since if "$pCO_2$" is replaced by "$pCO_{2,half}$", $GPP$ is not $GPP_{max}/2$ (though close to).

Yes, it's a bit strange. We note this internal inconsistency on lines 261-262.

- Eq. 19: To be fully consistent, please add commas (",") in fluxes variables: $F_{w,org}$, $F_{w,carb}$, $F_{b,org}$ and $F_{b,carb}$.

 Thanks, fixed.

**Supplement**

- Line 13: "$\psi$" is the southward transport (i.e., positive southward), not equatorward.

Thanks, corrected

- Line 16: "1.5x10^4" seems to be a typo ("x" instead of "\times"?). Please add also its units (J/kg). Besides, based on Siler et al. (2018), g should here be 0.06*$h_{eq}$, to match Eq. 8 of Siler et al. (2018).

Thanks, the 1.5x10^4 term is ~6% of modern $h_{eq}$, which we mischaracterized in the text. Our formulation effectively assumed that gross moist stability was constant with warming at the equator, and increased with warming to the north and south. However, we agree that gross moist stability should increase with warming even at the equator, following the Siler paper. The code and text have been updated so g depends on $h_{eq}$.

- Line 72: Missing reference to main text figure.

Corrected.

**Supplement tables (S1–S3)**

- Title of Table S3 is incorrect. It should be something like "carbon cycle and other geochemical parameters", not "weathering model".

Corrected.

- "$k_{reac}$" and "$r_{max}$" are a little confusing since they're both described as "reaction rate" but have different units, and do not represent the same thing (dissolution rate per unit of mineral surface VS dissolution rate per unit of water volume).

- What is "$\varepsilon$" in Table S3? How does it relate to "$\varepsilon_{b,org}$" and "$\varepsilon_{b,carb}$"? It seems that one value is missing to get the value of both "$\varepsilon_{b,org}$" and "$\varepsilon_{b,carb}$".

We assume epsilon,b,carb is zero (now stated in text and table S3).

* Missing parameters

- "$A$" (specific mineral surface area, from Eq. 12)
- "$m$" (molar mass, from Eq. 12)
- "$t_z$" (soil, or weathering zone, age) in equation 12 is named "$T_s$" in Table S2
- "$T_0$" (reference temperature in Eq. 13). It is not mentioned in the main text either.
- "$\delta^{13}C_{w,org}$"

* wrong or missing units

- "$c_P$" should be in J/kg/K, not J/kg
- "$R_v$" should be in J/kg/K, not J/kg
- "$\alpha$" should be in K^-1, not unitless
- "$C_{LW}$" should be in W/m2. not unitless
- "$B$" should be in W/m2/K, not unitless
- "$M$" should be in W/m2, not unitless
- "$L\varphi$" should be in m, not unitless
- the units of "$D$" is spelled "sec" instead of "s"

Thanks, the above corrections have been made.

**Review of "All aboard! Earth system investigations with the CH2O-CHOO TRAIN v1.0"**

by

Tyler Kukla, Daniel E. Ibarra, Kimberly V. Lau, and Jeremy K.C. Rugenstein

**Summary and general comments:**

Kukla and colleagues present the new computationally very efficient Earth system model framework, CH2O-CHOO TRAIN, that combines three existing modules: i) a zonal mean Moist Energy Balance Model (MEBM; after Roe et al., 2015), ii) a continental weathering module (after Maher and Chamberlain, 2014), and iii) a simple box model for the long-term carbon cycle (after Caves Rugenstein et al., 2019). The MEBM can be configured with different geographies to investigate its effect on temperature and runoff distributions for global weathering and carbon cycle dynamics. Especially due to the simple carbon cycle representation, the model simulates about 1 million model years in thirty minutes on a standard laptop. As such, it fills a gap between highly parameterized conceptual box models and Earth system Models of Intermediate Complexity (EMICs); thus, it represents a valuable new model for understanding interrelated Earth system dynamics.

Thank you for the thoughtful and thorough review!

The low computational demand of CH2O-CHOO TRAIN makes it very useful for large-ensemble experiments that are needed for uncertainty quantification. Another advantage the authors mention is that their model is "highly customizable", i.e., "making it easy to directly modify processes in the climate system". Unfortunately, both model strengths are not very well exploited in the manuscript. The model behavior is not compared for the different continental configurations, and only a few parameters are changed to lower and higher values. Why did the authors not put more work into investigating how the TRAIN really behaves, e.g., under different perturbation scenarios and/or model configurations (i.e., investigating structural uncertainties)? Many processes are not explicitly represented in the model and are thus highly parameterized and a large source of uncertainty. Therefore, a comprehensive sensitivity analysis is needed to quantify this uncertainty, identify the most sensitive parameters and adequately understand how the model works (see e.g., Pianosi et al., 2016, 2015, for some methods and ideas). Because this is the first CH2O-CHOO TRAIN model development paper and the model runs so fast, a comprehensive sensitivity study is needed and feasible.

Thank you for this concern. To address this, we conduct two sets of sensitivity simulations that allow us to quantify interaction effects between variables in the model (now Figure 5). We show that changing multiple inputs often leads to a climate response that differs from the sum of each individual change, and this is especially true when simulating low-CO2 conditions. These runs also emphasize how climate

responses to any variable are state-dependent, meaning the results of any sensitivity study may not apply to all use cases (we expand on this point later in the "Specific comments" section).

We agree that "customizability" is not the right word for the point we wanted to make. We have re-written this motivation around the model being flexible and making it easy to probe individual climatic processes that are hard-coded or emergent properties of other models.

Some parts of the manuscript are not well explained (motivation of the experiments) or information is missing (e.g., a comprehensive table stating model parameter names, values units, references; organic carbon burial & weathering; isotopic balance of the system) or confusing (e.g., is S and P calculated by the weathering module: if yes, how and why?).

Overall, I think the model can be a valuable tool to understanding Earth system dynamics and past climate variations which crucially depends on the combined use of different numerical representations of the Earth system. However, the manuscript would benefit from some more work to help the model uncertainty better and to showcase applications of the CH2O-CHOO TRAIN.

**Specific comments:**

Abstract: Please shorten some parts of the background and include a few main model results presented in the ms.

Thanks, done.

It could be made more clear what the main improvements of the model are compared to previous approaches, like COPSE or GEOCARB: I suppose it is the 1D atmospheric energy balance model and the improved water cycle? And also the possibility to represent a continental configuration for the MEBM model?

Yes, these are the primary distinctions. We clarify this point in lines 54-55.

Related to this: be more specific how the climate processes in the model can be modified. This is mentioned e.g. in the abstract, introduction (~line 45) and the start of Section 3 (line 369). The authors are the most specific when they mention "The model is designed to be highly customizable, making it easy to directly modify processes in the climate system such as ..." (line 44-45) and that the processes are either highly parameterized in simpler models or emerging properties in more complex models. I understand that this is a major strength of the model therefore it would be good to discuss this in more detail, give examples what specifically can be changed and how (I suppose the parameters changed in your experiments are not all possible parameters). A table summarizing what parameters can be changed to affect the different processes in what direction might be helpful.

We have refocused this part of the motivation around the model's ability to probe specific climate-carbon cycle interactions that are either absent from simpler models, or emergent in more complex ones. We did this recognizing that "customizability" is vague and probably distracts the reader/user from our motivation of building a model that better-represents basic climate-carbon cycle dynamics.

In the very beginning I thought the continental configuration would also impact the Long-term C-cycle but it actually does not. That should be made more clear in the text and in Fig 2.

We have added lines 281-283 to clarify this point and repeated this clarification in the caption of Figure 2. To be clear, the user may probe the effect of changing continental geography on the long-term C-cycle by initializing a new geography with the C-cycle fluxes of an old one and letting the model reach a new steady-state. Geography also has a small effect on the strength of the weathering feedback via equation 21.

**Figure 1:**

It was not obvious to me how some components shown are necessary to initialize the model. E.g., it is unclear how knowledge about lithology, soil age, soil pCO2 influences the weathering model (I am not able to judge the MEBM part – this is not my expertise). I did not see parameters for them in the main manuscript. E.g. soil age (TS) is given in Table S2 but I can't see an equation where it is used – or it is twz in equation (12).

Thanks for pointing out this issue – we have modified the text boxes in Fig. 1 to use the same terms we use in the "Weathering" section of the model formulation (substituting "soil" for "weathering zone").

**Model**

The main manuscript does not include a table stating the main model parameters, their values, units and references. This makes assessing the model construction difficult. Some are given in the SI – this could go into the main manuscript. But do the Tables S1 – 3 include all important model parameters? Please make sure the units for all model outputs are stated, e.g. no units are given for qland and Fw,sil,carb after equations 8 + 18 or what is meant by [C] (+ units) in line 195 is not 100% clear.

Thanks, done.

Also, I suppose various model parameters are very uncertain. Therefore, a sensitivity study which assess the importance of these parameters for different model outputs and calculates quantitative sensitive indices would be very informative in order to understand how the model behaves and its results. In line367 – 368 you say: "These experiments are not meant to be an exhaustive sensitivity analysis of the model." But this is the first version of the model, therefore, in my opinion, a more complete sensitivity analysis is necessary. In contrast, the often long and sometimes vague discussion of model limitations (Section 2.5 & 5.1) could be shortened.

We added the factorial experiments mentioned earlier to address non-linear responses to perturbing multiple inputs. This is the most relevant sensitivity test that we can run for a couple reasons. First, the model is a patchwork of existing frameworks that we stitched together, and previous work has already addressed the sensitivity of these individual components. Beyond this cited work, our motivation here is to demonstrate the sensitivity of model results to climate-carbon cycle interactions that were not captured by any previous individual framework. This leads to the second reason—the model response to perturbing any variable is highly state-dependent. The magnitude and (in some cases) direction of the climate response to a perturbation depends on the glacial conditions, continentality, climate sensitivity, and even the values of the other variables (as we show). As a result, we think it's more useful to show the reader/user why the model responses are so complex rather than try to quantify model sensitivity knowing any answer will not broadly apply. In this case, the complicated model responses come from interaction effects between variables, as well as differences in the spatial pattern of temperature and hydroclimate change to $CO_2$, relative to continentality. This conceptual framework is what we lay out in the paper, and what we think is most useful for understanding how the model will respond to a given change for a wide range of conditions.

Simulating modern conditions:

It would be very informative to provide a more in depth evaluation of the model results with modern boundary conditions – which, I think, should be the main reference simulation to establish that the model works well. E.g., you could compare your model output with other estimates and/or observations (e.g., plot this in Fig. 4 A – C and also the simulated E minus P). Do I understand it correctly that the output of the C-cycle model is prescribed for the steady-state condition, i.e. all the initial fluxes in Table S3?

Thanks, we considered this as well, but we are not so sure that an in-depth evaluation of the modern would be very helpful.

The climate component of the model was thoroughly tested against modern conditions by Siler et al. 2018, and the long-term carbon cycle (the other component of the model, including weathering) is informed by fluxes that are, themselves, informed by data—a comparison wouldn't be so meaningful. Zonal-mean weathering patterns could be compared to data, but the comparison is far from straightforward because of challenges with uncertain modern patterns of silicate weathering and with deriving a zonal-mean map from incomplete modern data.

Yes, the steady state initial condition is prescribed. Changing something in the model (volcanism, atmospheric energy transport diffusivity, continentality, etc) will generally create a carbon cycle imbalance that will push the model to a new, not prescribed, steady state.

**2.1 Moist Energy Balance Mode**l – I don't have the expertise to review this in detail.

But where does the temperature threshold of -5°C for the appearance of ice come from? And should this not be very different for ice sheets (on land) and the formation of sea ice? Please give a reference and justify.

We added a reference for justification (North 1981). We ignore the distinction between land and sea ice in this version of the model, but select a higher temperature threshold (most modelers go with -10C) to account for the warmer conditions where land ice might exist.

How is temperature calculated in the MEBM?

Temperature is solved from equation 4. The MEBM simulates the zonal profile of moist static energy, which can be converted to temperature given some relative humidity.

What happens to terrestrial runoff over the ice sheets? Should this not be significantly reduced because the precipitation is snow and becomes part of the ice sheet? This might not be important for the default setup of the model because it does not affect weathering but for your experiments "effect of ice cover on weathering" I suppose it is.

Indeed, this is the point of our "effect of ice cover on weathering" simulations (Fig. 6). To clarify how "effective runoff" differs from P minus E output by the MEBM, we modified Figure 3 to P minus E over the ocean (dashed lines) and "effective runoff" over land (solid lines). As discussed in Sect 4.4, the impact of ice on weathering is a tunable parameter. Section 4.4 and Fig. 6 is a sensitivity test with an extreme case (Northland) showing widely different climate responses to the same volcanic perturbation depending on how much weathering is permitted beneath the ice sheets.

**2.2    Weathering:**

lines 239 ff. "We parameterize maximum carbonate weathering reaction rates as being 1000 times faster than silicate weathering… " Is this a reasonable assumption? I would assume these parameters have large uncertainties and a large effect on the model output. I would suggest include these (and similar parameters) in a sensitivity study.

Please provide information for how organic carbon weathering is calculated? Is it constant?

With the help of these reviews, we found issues with our weathering formulation and have corrected these parameters and the code. This involves a new scaling for carbonate weathering defined in lines 263-265. Thanks for pointing out the confusion with Fworg – we now clarify in lines 380-385 that it is constant, and discuss the implications.

In the text (line 101) and in Fig. 1 the authors mention that the weathering module calculates fluxes of P & S, but this is not discussed/introduced in the model description. Why is this even in the model? If I understand it correctly, your ocean PP is not simulated (Corg burial is only scaled to CaCO3 burial) – hence P is not needed. And I don't understand why a S-cycle would be needed? There is no information in the manuscript!

Apologies for this confusion – This is a relict of an older model version. In the version presented in this paper we do not explicitly capture P and S cycling. The text and figure have been corrected.

This section should also include details for the isotopic signals of the weathered Corg and carbonates – some of it is given in Table S3 but it would be good to include it in the main manuscript.

Thanks, we have added these details to lines 288-290 in the carbon cycle section in an effort to streamline the isotope discussion.

Table S3 does not state the initial silicate weathering rate. I suppose it is equal Fvolc?

Thanks for catching this – we added it to Table S3.

**2.3    Carbon cycle:**

Please state how organic carbon burial is scaled to CaCO3 burial? Is it done as by Kump & Archer (1999) to isotopically balance the system?

This is now clarified in lines 299-301.

Steady-state of the model:

Related to the last comment: To achieve steady-state is it also necessary that organic carbon burial equals organic carbon weathering – or is this always the case in the model? Or do not have the option to restore some of the buried Corg via volcanic outgassing (see e.g., Lenton et al., 2018, Table 2)?

Our formulation allows organic carbon weathering and burial to exist be imbalanced in certain conditions. We now make this clear and discuss the implications in lines 383-385.

Please give information how the system is isotopically balanced! This is not discussed.

We now clarify how we achieve isotopic mass balance on line 290.

**2.4    Coupled climate-carbon cycle model initialization and integration**

Line 299: what does 'the carbonate speciation described above' mean? I suppose, you recalculate it every timestep, right – and also Wcarb,sil? Is pH also updated? How?

We replaced "above" with "in the previous section". We also clarified that W is only computed at initialization, then held constant (line 268). pH is updated using the same carbonate speciation approach that we discuss in the "Carbon Cycle" section (these equations are well-described in the cited literature), though to clarify this point we added that carbonate speciation is computed at each timestep (lines 337-339).

Line 300: What is meant by "temperature guesses"? Why do you need to guess? Also the rest of the paragraph is unclear to me – why does the model result in a snowball? Does that mean the temperature is everywhere below the threshold of -5°C? And what does "In this case," (line 301) refer to?

Thanks for drawing our attention to this confusion, we corrected inaccurate text about updating the temperature guesses based on the MEBM result (in this paper, the temperature guesses are constant through time, consistent with how they're described in section 2.1.4). This involved deleting the confusing "in this case" reference.

The temperature guesses are described in lines 210-213 (we now redirect the reader back to this section on line 329). This is a standard method for solving certain differential equations such as boundary value problems – you start with a "guess" about what the solution should be, and the numerical solver uses it as a starting point to iteratively solve for the best solution.

We clarified that low temperature guesses and low $pCO_2$ can lead the model to find a snowball climate (all temperatures below -5deg) as the most stable state (lines 329-330), though small perturbations to the initial temperature guess quickly melt the snowball, indicating that the result is not robust (and it shouldn't be, given the relatively high $pCO_2$ levels we simulate compared to what should drive a true snowball).

**2.5     Model assumptions and limitations**

This part, for the most part, speaks about model limitations. And could therefore be removed from Section "2 Model Formulation" and go to the end of the manuscript and combined with "5.1 Model applications and limitations"

We agree and have moved parts of the earlier section to 5.1,2. We also changed the titles of these sections. Sect 2.5 is now model assumptions (the limitations are discussed outright), and 5 is model applications and their limitations.

**3     Model Experiments**

The different experiments are not very well motivated or explained. This could be expanded and subsections could be used as in Section 4. A table summarizing the setup of the different experiments would be very helpful.

We added to (and clarified) our motivation on lines 388-390.

Please state clearly what the background climate & C-cycle state is for set of experiments?

For Fig. 6 – 8 only the normalized values are given – without information how they are normalized.

We added the initial Co2 conditions to Figure 2, outlining the climate forcing for each set of simulations – background carbon cycle is the same as defined in section 2. We also describe how the results are normalized in lines 520-522 and the figure captions.

Would it not be informative to do the same perturbation experiment with every continental configuration to evaluate the effect of geography?

Perhaps, but we think that amount of information would distract from our goal of demonstrating why geography matters. We select geographic configurations that capture the effect of spatially variable climate responses – since each climate perturbation has a different spatial pattern, we use different geographies.

Especially, the description of the second series of experiments using the "Northland" geography is unclear. What parameter are the authors changing? And what is the effect: How much weathering happens under an ice-sheet? Or also the amount of runoff? This is not clear from Section 3. Maybe that is why I struggle to understand the model results (see comments on Section 4.3).

We added text to clarify this at lines 412-422. This term was not explicitly written in the earlier draft, but we now have it as k_ice in the new manuscript (see equation 10).

Subtropical continents:

I am not entirely sure how the last set of experiments links to the hypotheses of the breakdown of the silicate weathering feedback. You arbitrarily change the runoff in an experiment that is at steady-state which then causes the feedbacks to respond. And yes, you get a runaway greenhouse for a small continent where the runoff is very sensitive to changes in w and the stabilizing strength of the silicate weathering is weak. Would it not be better to to setup different steady-state experiments with the three configurations (and different values of w) and then perturb the system (i.e., by injection of CO2) and see how the responses are different?

We removed this experiment from the revised text because we agree with both reviewers that it is not very useful. We wanted to demonstrate that our model does not enforce a negative silicate weathering feedback—the negative feedback is an emergent property of the global climate response to CO2. Though this point is probably made clear already in the text, and the unrealistic nature of this special case is more distracting than helpful.

**4    Results**

**4.1    Reference Simulation**

Here you talk again about the temperature guess. This is unclear to me.

We clarified how this works in section 2.1.4 and again in section 2.4 (as noted previously in our responses).

Line 424-426: "If both temperature guesses are the same, the first pole to glaciate in the meridionally symmetric case will depend on the tuning of the numerical solver."

Can the authors please give more details here. Maybe this is related to my question how temperature is calculated in the MEBM?

Done. Basically, it depends on which pole the numerical solver wrestles with first (i.e. whether the south pole is the "left guess" (the first one) or the "right guess").

**4.2    Response to abrupt pCO2 increase with modern geography**

What is the isotopic signal of the carbon injection?

-20‰, now stated on line 401.

The motivation given for this experiment is (line 376): "This simulation is used as a verification of the coupled model's performance in comparison with other, similar simulations across the model hierarchy." This is not done at all.

Thanks for catching this – we clarified the motivation as a basic comparison (line 402) and we compare the response timescale to modeling results and data (line 476).

There could be more content in this section. The model runs so fast. Why do the authors not put more work into investigating how it behaves under different perturbation scenarios or model configurations or for different parameter settings. I feel like this is a missed opportunity to understand their model better.

Our motivations for constructing this model and presenting this work are different than assumed by this comment. We view the spatial dimension and the climate system's complexity and flexibility as the model's key advantages. In the discussion we explain why directly comparing factors such as the recovery timescale across different configurations creates an "apples-to-oranges" problem. Any time you change something that will impact the steady state carbon cycle (including a climate variable, continentality, etc), you implicitly change the strength of the silicate weathering feedback. One can't really test, for example, whether a low-atmospheric energy transport efficiency world is more resilient to perturbations than a high-AET efficiency world—the change in perturbation response can't be deconvolved from the required change in W (equations 25, 26), or in volcanic emissions, or in the baseline climate state (all of which add confounding factors to the comparison). Depending on what it is about the model one wishes to understand, it's not clear to us that repeating this experiment under different conditions will inform more general cases than the one presented here.

Unclear how the last two sentences of the section fit in.

Agreed – we deleted them.

**4.3    Varying the effect of ice cover on weathering**

If I understand the experiment correctly two parameters are changed at the same time:

1.      % of effective runoff

2.      volcanic outgassing

I think this makes it difficult to disentangle what's going on. Imagine, keeping volcanic outgassing unchanged and only changing the effective runoff: this should already result in a different equilibrium climate because some weathering is possible under the ice sheets. If I am correct, this should be evaluated first by the authors.

This is a helpful point that gets at the challenge of isolating any individual variable's effect on the climate response to a volcanic perturbation (for example, see our response above regarding why we didn't expand the 5000Pg perturbation section). Changes in nearly every climate variable require other changes (either in volcanism, W, or $CO_2$) to balance the C-cycle.

However, in this specific case we don't face this problem—we now explain why in lines 512-517. "We note that changes in $k$ice have no impact on climate or the long-term carbon cycle in an ice-free world (where $k$ice = 1 everywhere). This is why all simulations start at the same ice-free initial conditions in Figure 6. As a result, these simulations can be directly compared because all terms that are defined when the model is initialized—including $W$sil which impacts the strength of the silicate weathering feedback separately from $k$ice—are equal. If we initialized the model in a glaciated state, then $W$sil must vary with $k$ice to maintain a balanced carbon cycle at the first timestep, and our results would confound the direct effect of changing $k$ice plus the indirect effect of changes in $W$sil".

I find the text difficult to follow. In general, it might be good to start from your default setup: which is, as I understand it, is 0% effective runoff. And then describe what happens if effective runoff is increased.

Hopefully our added text addressing the point above clarifies this issue (increasing effective runoff has no effect on climate or carbon cycling in an ice-free world).

Please plot the global area of ice-cover as it is talked about in the text, it's a main part of the experiment and important to understand what's going on. This might also solve my confusion with the statement in Fig. 5 "Ice sheet growth limits runoff": Why would this be largest in the experiment with the warmest climate? Ice sheets should expand the least here.

Added. We re-wrote parts of the first paragraph of section 4.4 to be clearer – ice sheet growth limits runoff more when effective runoff % is low. This effect wins out over the warmer effect, but with some dampening (as we now explicitly state).

Also what about the ice-albedo-temperature feedback? Does it not play a role for the results of Fig. 5? It is only mentioned at the end to explain the step-changes.

Sure, the ice-albedo feedback plays a small role by decreasing temperature locally, but the feedback itself is a feature of the system that remains constant in all simulations. That is, the strength of the

feedback (which depends on how different ice albedo is from ice-free albedo) is held constant in much the same way that the warming effect of the water vapor feedback is also held constant.

Why does the model calculate higher mean runoff in the 100% effective runoff setup which is the colder climate state. I had the impression runoff scales mainly with temperature (see, e.g. Fig. 4E, F).

We added clarifying text to the figure caption. This figure panel shows the runoff relevant for weathering (that is, it accounts for changes in effective runoff).

**4.4      Instantaneous change in moisture recycling efficiency**

Why are all results now shown normalized and how is this done?

They're normalized because we are now comparing different continental configurations which yield different initial climate states and different global temperatures (though all simulations maintain a greenhouse climate). We added text to explain this in the figures and line 520.

Fig. 6 D: Why is there first an increase in net C emissions? This is never discussed.

Thanks, now discussed in line 523-524.

In my opinion, the last paragraph (lines 494 - 502) does not belong into the results section. If at all it could into the limitations section.

We agree. We deleted this paragraph.

**4.6 Subtropical continents**

line 530: "Runoff tends to decrease with warming in the subtropics in the MEBM module" can you show model output for that? It is in contrast to the statement in lines 436 – 437: "runoff is generally insensitive to global climate between 30 and 50 degrees latitude".

We deleted this section, but we don't see much of a contradiction. 30-50 degrees latitude is better described as the mid-latitudes, not the subtropics (we center our subtropic geographies around 10-30 degrees).

The description of Fig. 9 is very confusing as it jumps back and forth between increased runoff and decreased runoff experiments and explains how runoff changes through the transient experiments.

We agree. Section deleted.

Isn't the small continent of business belt world a big factor why it runs into a runaway greenhouse: i.e. the weathering feedback is weaker than in the other worlds and runoff is probably also more sensitive to changes in w.

For the sake of discussion, no the low land area is not a factor in why it runs away. In fact, the low land area implicitly yields a stronger weathering feedback, rather than a weaker one, by increasing the Wsil required to balance the carbon cycle. The location of the land makes the feedback weak (and positive).

**5        Discussion**

It is not really a Discussion paragraph. Maybe something like "Scope of applicability and limitations" would fit better.

We agree. We deleted section 5.1 and moved the title to what was previously "Discussion".

The numbering is a bit odd. Why do you need 5.1 if there is not a 5.2?

Thanks, fixed.

Maybe move "2.5 Model assumptions and limitations" to the end of the manuscript and combine with information given in "5.1 Model applications and limitations". Or it might be useful to have two different sections: 1) Model applications 2) Model limitations

Thanks, we agree and have made changes – see our response to this point when it was first mentioned earlier in the response document.

**Conclusions**

The manuscript is missing a conclusion paragraph.

Because the paper is intended to highlight a new model and its distinctions, we view section 6 as an appropriate way to conclude the paper by highlighting future work possible with relatively straight-forward improvements to the model.

**Technical corrections:**

Line: 34: 'the most physically realistic' maybe better write 'a more mechanistic'

Thanks, changed.

line 43: add: and 'a' box model for …

Done.

line 65: Maybe '… account for more spatial dynamics than the 0-D representations …' because it is just 1D so does not account for all the spatial dynamics

Thanks, we added "(at least meridionally)".

line 68: "precipitation (assumed proportional to runoff)" The phrasing is a bit unclear to me.

Do you mean runoff is proportional to precipitation (because it sounds like the model calculates precip first)? And why is this needed here? Because you are eventually interested in runoff?

We deleted the parenthetical and added that they do not capture runoff at all in their model. The distinction is important because runoff is more directly related to weathering than precipitation. By not simulating runoff, they assume that changes in precipitation are mapped directly onto changes in runoff (effectively ignoring changes in E).

Line 73: maybe say 'in a more mechanistic way'

Done.

line 97: delete the first occurrence of the word 'box'

Done.

line 100: add 'long-term carbon cycle model' + singular for 'outputs'

Done.

line 101:should read: These fluxes are used to calculate

Done.

Equation (7) w is not defined here. In Figure 6 you call it evapotranspiration. Which is confusing because ET was defined as evapotransipiration after Eq. (7).

We changed figure 7's caption to state that we're looking at a change in evapotranspiration efficiency.

Equations: be consistent with commata in subscripts, e.g., compare (19) vs (21)

Thanks – fixed.

line 195: It might be good to explicitly state here: "We calculate solute concentrations for inorganic carbon, [C], … " To make clear that all forms are considered here.

Thanks, added.

line 281: please provide a reference for the 10° colder also in the text – not just in Table S3

Thanks, added.

line 363: small letter p in phosphorus

Fixed.

Missing references for PETM C injection: lines 375-376, 449-450

Thanks, added.

line 420: why geography? You are only looking at the cat-eye geography here!

Just reminding the reader that this $CO_2$ threshold for glaciation should not be taken as a generalizable threshold (i.e., this result is not comparable to what we'd expect for a modern Earth).

Line 471-473: this is trivial

Perhaps for the k_ice experiment, but it's relevant context for understanding that k_ice doesn't matter for some of our other simulations (such as when land is in the tropics).

Fig. 6: And related text, define what net C emissions are. The names of the geographies are different in Fig. 6.

Fixed – net C emissions now defined on line 519-520.

line 636 + 637: Please be careful: GENIE (i.e., Holden et al., 2016; Ridgwell et al., 2007) does not include a representation for ice sheets but only a sea ice model.

Thanks, now clarified on line 631-632.

Fig. S4: has the wrong exponents for the D values

Thanks for catching this – fixed.

**References**

Caves Rugenstein, J. K., Ibarra, D. E., and von Blanckenburg, F.: Neogene Cooling Driven by Land Surface Reactivity Rather than Increased Weathering Fluxes, Nature, 571, 99–102, https://doi.org/10.1038/s41586-019-1332-y, 2019.

Maher, K. and Chamberlain, C. P.: Hydrologic Regulation of Chemical Weathering and the Geologic Carbon Cycle., Science (New York, N.Y.), 343, 1502–4, https://doi.org/10.1126/science.1250770, 2014.

Pianosi, F., Sarrazin, F., and Wagener, T.: A Matlab toolbox for Global Sensitivity Analysis, Environ. Modell. Softw., 70, 80–85, https://doi.org/10.1016/j.envsoft.2015.04.009, 2015.

Pianosi, F., Beven, K., Freer, J., Hall, J. W., Rougier, J., Stephenson, D. B., and Wagener, T.: Sensitivity analysis of environmental models: A systematic review with practical workflow, Environ. Modell.

Softw., 79, 214–232, https://doi.org/10.1016/j.envsoft.2016.02.008, 2016.

Roe, G. H., Feldl, N., Armour, K. C., Hwang, Y.-t., and Frierson, D. M. W.: The Remote Impacts of Climate Feedbacks on Regional Climate Predictability, Nature Geoscience, 8, https://doi.org/10.1038/NGEO2346, 2015.

---

## Author Response (AR2)

**Response to reviewers**

The authors have properly addressed all the comments I had concerning the first submission, and considered the corrections I suggested. The reorganization of the manuscript, mostly following the comments of reviewer #2, is relevant. So far, I don't have any more request on this revised version, save a few minor corrections. I do have a last broad question on the model's behavior, but I don't consider it requires to modify the current manuscript.

I was startled, looking at Fig. 7 (former Fig. 6) to see a complete latitude flip in the sensitivity of runoff to the "omega" exponent. As far as I understand, the only changes that were made in the MEBM code were to replace "q" by "q/rh" in eq. 8, and to reformulate the gross moist stability from "h(0) - h(x) + cst" to "1.06*h(0) - h(x)". Are those two changes enough to explain the difference? We're talking about a 5% versus a 25% of runoff increase when reducing "omega" in Polarslice experiment. The author's justification is that the ratio E0/P was closer to 1 at high latitude than in the tropics, in the former version of the model, but is now closer to 1 in the tropics. This is not improbable per se, but it reveals quite a high sensitivity of the model to these parameterizations.

Thanks for pointing this out. We ran some sensitivity tests and couldn't reproduce the change. Ultimately, we traced it back to an error in the first submission that was fixed when we re-ran our simulations for the revisions. In the first submission, we accidentally plotted the results for a mid-latitude belt rather than a polar belt. This is why omega was closer to one in "polarslice" (actually mid-latitude slice) world and now is closer in "tropicslice". We re-ran polarslice with the old code and got results very similar to the new code, as expected. We have also gone back and confirmed that the rest of our plots match the stated geography.

Minor corrections:

The modifications of Fig. 4 ("% of global discharge 320 ppmv", weathering units, and indication "for each equal-area latitudinal grid cell" in the caption) are welcome, but should also be made on Supp. Fig. 4.

Thanks, these changes have been made.

Fig. 7's caption needs to be updated: the "The slower and larger magnitude climate response" is now of Tropicslice, comparred to Polarslice.

Thanks, changed.

Similarly, Supp. Fig. 3's caption needs to be updated: both configuration now have similar runoff sensitivity to temperature, and Polarslice has a larger weathering sensitivity to pCO2.

Thanks, changed.

Line 223: The Damköhler weathering coefficient has the dimension of a runoff (unlike the Damköhler number, that is dimensionless).

Correct, thanks for catching this. Changed.

Line 312 and in Supp. Table 3: the values of [Ca], [Mg] and [SO4] are in mol/m3, not in mol/L

Thanks, values have been changed to mol/L in both places.

The parameter "theta" (Clausius-Clapeyron coefficient) is still referred as "alpha" in Supp. Table 1.

Thanks, fixed.

The reference temperature "T0" in Supp. Table 2 is probably 14°C, not 14 K.

Indeed! Thanks.

I am pleased that Kukla and colleagues paid much attention to the reviewer's comments. They addressed all comments and questions, and adjusted the text where appropriate. They corrected bugs in the model code in response to two reviewer comments and re-ran all experiments.

The authors also added a section including some sensitivity experiments to address my main comment of the first review. I thank the authors for this. However, I am a little disappointed by this section because it is not a comprehensive sensitivity analysis and therefore does not really address what I had in mind with the comment: "Many processes are not explicitly represented in the model and are thus highly parameterized and a large source of uncertainty. Therefore, a comprehensive sensitivity analysis is needed to quantify this uncertainty, identify the most sensitive parameters and adequately understand how the model works."

The new results only show that interacting effects between model parameters exist (which is to be expected) without quantifying, for instance, which parameters have more significant interacting effects than others, or to which model parameters model results are most sensitive.

I appreciate that "the model is a patchwork of existing frameworks that we stitched together, and previous work has already addressed the sensitivity of these individual components." However, from coupling different modules together, modelers have experienced that a coupled, interactive system can react differently than individual components alone. I also understand that the model behavior depends on the boundary conditions. Therefore the author's approach of simulating a low and a high pCO2 world using a modern continental configuration (Section 4.3) is a good way to provide some general guidance but could be extended.

Overall, I had hoped for a more elaborate quantitative analysis of some largely conceptual/abstract parameters (i.e., without explicit links to physics, biology, or chemistry) in order to provide a better understanding of the model's considerable parameter uncertainty. Methods for quantitative sensitivity analysis exist (see, e.g., the Elementary Effects Tests in the publications I provided in the first round of reviews) and could be quickly adopted, especially when considering the low computational costs of the model.

However, if the authors and the editor decide that the current sensitivity analysis is sufficient for convincing readers of the conceptual model framework, I will not stand in the way of publishing the manuscript (once the minor comments below have been addressed). As I said before, the authors did a really good job in setting up this new Earth system model framework which fills the gap between conceptual box models and Earth system models of Intermediate Complexity.

Specific comments (only minor comments):

Line 188: kice: technically you need to distinguish between kice and kland with kland=1 always (or in a similar manner). Otherwise, it looks like you are changing the size of the effective runoff also over ice-free areas.

We added a piecewise equation (eq. 11) to formalize the text saying that kice=1 over ice-free land, and modify the text to clarify that we test the sensitivity of kice over glaciated regions (holding kice over land constant).

Lines: 264 "and carbonate [C]sil,eq 2 times greater" I find this difficult to understand - please rephrase.

Thanks, we revised the text to clarify that max equilibrium carbonate concentrations are two times greater than silicate.

Line 413+414: I think this should be equation 10, right? There is no kice in equation 9.

Thanks for catching this. We changed to equation 10 (and also refer to the added eq. 11 for completeness).

line 415+416: "For example, in Figure 3B, kice is zero, which is why the solid lines go to zero

at glaciated latitudes."

Please delete this comparison to a different model setup (Fig. 3 uses cat-eye geography and different pCO2). It is confusing that you suddenly refer to a different configuration to explain the kice parameter. If you want to give more general information on kice – around line 190 would be a better place.

Thanks, deleted.

Line 416: "Here, we test the model response to varying kice from 0 to 1 (in all other simulations, kice is set to 0)."

This sounds like you change kice "from 0 to 1". Do you mean varying kice between 0 and 1?

Yes, we replaced "from 0 to 1" with "between 0 and 1".

3 Model Experiments:

Third set of experiments – halving of volcanic flux for different kice values

I misunderstood the experiment setup in the 1st version – two things were unclear from the text (to me):

1) that the initial state was ice-free

2) that kice is kept constant during the experiments. I thought kice was changed from the default value

simultaneously with halving volcanic outgassing.

I only understood this from reading your response – in particular the last part (lines 512 - 517). It would be good to include this information to "3 Model Experiments" as it describes the experiment setup and not the results. (Also thanks for including the iceline subfigure to Fig. 6.)

Thanks, we see how this was confusing and have clarified the model setup in section 3. The text now includes "Starting from an ice-free climate for each of the five glaciated kice values,… . The different kice values have no impact on climate until the decrease in volcanism is sufficient for glaciers to form." (lines 422-424).

Section 4.2: I still think this Section is a little low on content and could be improved. But that might be a personal preference and I am okay if the authors decide to keep the section as it is.

We appreciate the point, though this section is primarily meant to show the model captures a basic perturbation. We don't want it to be lengthy.

Section 4.3: Please define ΔT in Fig. 5 (is it ΔT of global mean surface temperature or does it take spatial differences into account)?

ΔT is global mean surface temperature change relative to the expected change if variables added linearly. Changes in surface temperature are not spatially uniform, so ΔT is sensitive to regional effects. We clarify that ΔT refers to global mean surface temperature in the caption of fig. 5.

Figure 6: I was still confused why "Ice sheet growth limits runoff". I think the note you added to the caption of Fig. 6 ("Note that (D) refers to terrestrial runoff relevant for weathering (which accounts for changes k ice ).") is very important. So Fig. 6D is not "Global mean runoff" but "Global mean effective runoff"? If yes, I do understand Fig. 6 and please change the y-axis label accordingly.

It is effective runoff, we've changed the y-axis and caption to reflect this.

---

## Author Response (AR3)

Per our email correspondence (copied below for reference) we have not made any changes from our previous upload.

-----Original Message-----
From: tykukla@uw.edu <tykukla@uw.edu>
Sent: Monday, August 21, 2023 4:28 PM
To: Copernicus Publications Editorial Support <editorial@copernicus.org>; tykukla@colostate.edu
Cc: jeremy.caves@gmail.com; editor@mailarchive.copernicus.org
Subject: RE: egusphere-2022-1000 (author) - manuscript needs minor revisions

To whom it may concern:

It appears the only minor revisions refer to Polina Shvedko's concerns on the file validation step. I believe I addressed these concerns in my last submission. Could you please clarify where I went wrong?

The first concern involves renaming the supplement. I deleted the supplement title, authors, and contact information following the instructions in the provided link, which state: "Supplements will receive a title page added during the publication process including title ("Supplement of"), authors, and the correspondence email. Therefore, please avoid providing this information in the supplement."

The second concern states: "Please ensure that the colour schemes used in your maps and charts allow readers with colour vision deficiencies to correctly interpret your findings". I have done this too. The only figure where colors may be difficult to distinguish is Figure 2, but in this case, color is aesthetic only. It is *not necessary* to interpret the figure. All colors required for interpretation are interpretable to readers with color vision deficiencies based on my Coblis uploads.

Best,

-Tyler Kukla

-----GMD Reply #1-----
Dear Tyler Kukla,

Thank you very much for your email.

I will get in touch with the handling editor to clarify what the needed minor revisions are and if they indeed refer to Polina's comments. I will get back to you as soon as possible.

Best regards,

Natascha
* * *
Copernicus Publications
The Innovative Open-Access Publisher

Natascha Töpfer
Editorial Support | Team Coordinator

-----GMD Reply #2-----
Dear Natascha, and to Tyler Kukla,

I obviously made some confusion between manuscript versions, and probably also with the state of other papers I handle as editor.

The last version of Kukla et al. is indeed now ready for publication, and we can now go further in the process.

I sincerely apologise for this mess, and once again thanks the authors for publishing in GMD.

Best regards,

Olivier Marti

@Natascha : I can not do nothing on the web site to change the paper state.